# Dynamic action of the Sec machinery during initiation, protein translocation and termination

**Tomas Fessl[1,2,3], Daniel Watkins[4], Peter Oatley[1,5†], William John Allen[4], Robin Adam Corey[4], Jim Horne[1,2], Steve A Baldwin[1,5‡], Sheena E Radford[1,2], Ian Collinson[4]\*, Roman Tuma[1,2,3]\***

[1]Astbury Centre for Structural Molecular Biology, University of Leeds, Leeds, United Kingdom; [2]School of Molecular and Cellular Biology, Faculty of Biological Sciences, University of Leeds, Leeds, United Kingdom; [3]Faculty of Science, University of South Bohemia, Ceske Budejovice, Czech Republic; [4]School of Biochemistry, University of Bristol, Bristol, United Kingdom; [5]School of Biomedical Sciences, Faculty of Biological Sciences, University of Leeds, Leeds, United Kingdom

**Abstract** Protein translocation across cell membranes is a ubiquitous process required for protein secretion and membrane protein insertion. In bacteria, this is mostly mediated by the conserved SecYEG complex, driven through rounds of ATP hydrolysis by the cytoplasmic SecA, and the trans-membrane proton motive force. We have used single molecule techniques to explore SecY pore dynamics on multiple timescales in order to dissect the complex reaction pathway. The results show that SecA, both the signal sequence and mature components of the pre-protein, and ATP hydrolysis each have important and specific roles in channel unlocking, opening and priming for transport. After channel opening, translocation proceeds in two phases: a slow phase independent of substrate length, and a length-dependent transport phase with an intrinsic translocation rate of ~40 amino acids per second for the proOmpA substrate. Broad translocation rate distributions reflect the stochastic nature of polypeptide transport.
DOI: https://doi.org/10.7554/eLife.35112.001

**\*For correspondence:**
ian.collinson@bristol.ac.uk (IC);
r.tuma@leeds.ac.uk (RT)

**Present address:** [†]Department of Molecular Biology and Biotechnology, University of Sheffield, Sheffield, United Kingdom

[‡]Deceased

**Competing interests:** The authors declare that no competing interests exist.

## Introduction

Protein secretion is essential for life; responsible for the delivery of proteins to and across the cell surface. The major route for this process is by way of the ubiquitous Sec machinery, comprising at its core a heterotrimeric complex: SecYEG in the plasma membrane of bacteria and archaea, and Sec61αβγ in the eukaryotic endoplasmic reticulum (ER). Pre-proteins are targeted to the Sec machinery with the aid of an N-terminal signal sequence (SS) or a trans-membrane helix (TMH), and translocated through the Sec machinery in an unfolded conformation (*Arkowitz et al., 1993*). This can occur either during their synthesis (co-translationally), or afterwards (post-translationally); in the latter case, pre-proteins are prevented from folding by cytosolic chaperones, such as SecB in bacteria (*Kumamoto and Beckwith, 1983*; *Weiss et al., 1988*). Bacterial inner membrane proteins are generally secreted co-translationally, while proteins destined for the periplasm, outer membrane or the extra-cellular medium, tend to follow the post-translational route, driven there by the ATPase SecA (*Hartl et al., 1990*; *Brundage et al., 1990*; *Lill et al., 1989*).

The protein-channel is formed through the centre of SecY, between two pseudo-symmetrical halves, each containing five TMHs (*Figure 1*) (*Cannon et al., 2005*; *Van den Berg et al., 2004*). When at rest the channel is kept closed by a short, usually α-helical plug and a ring of six hydrophobic residues, which serves to prevent ion leakage and dissipation of the proton motive force (PMF)

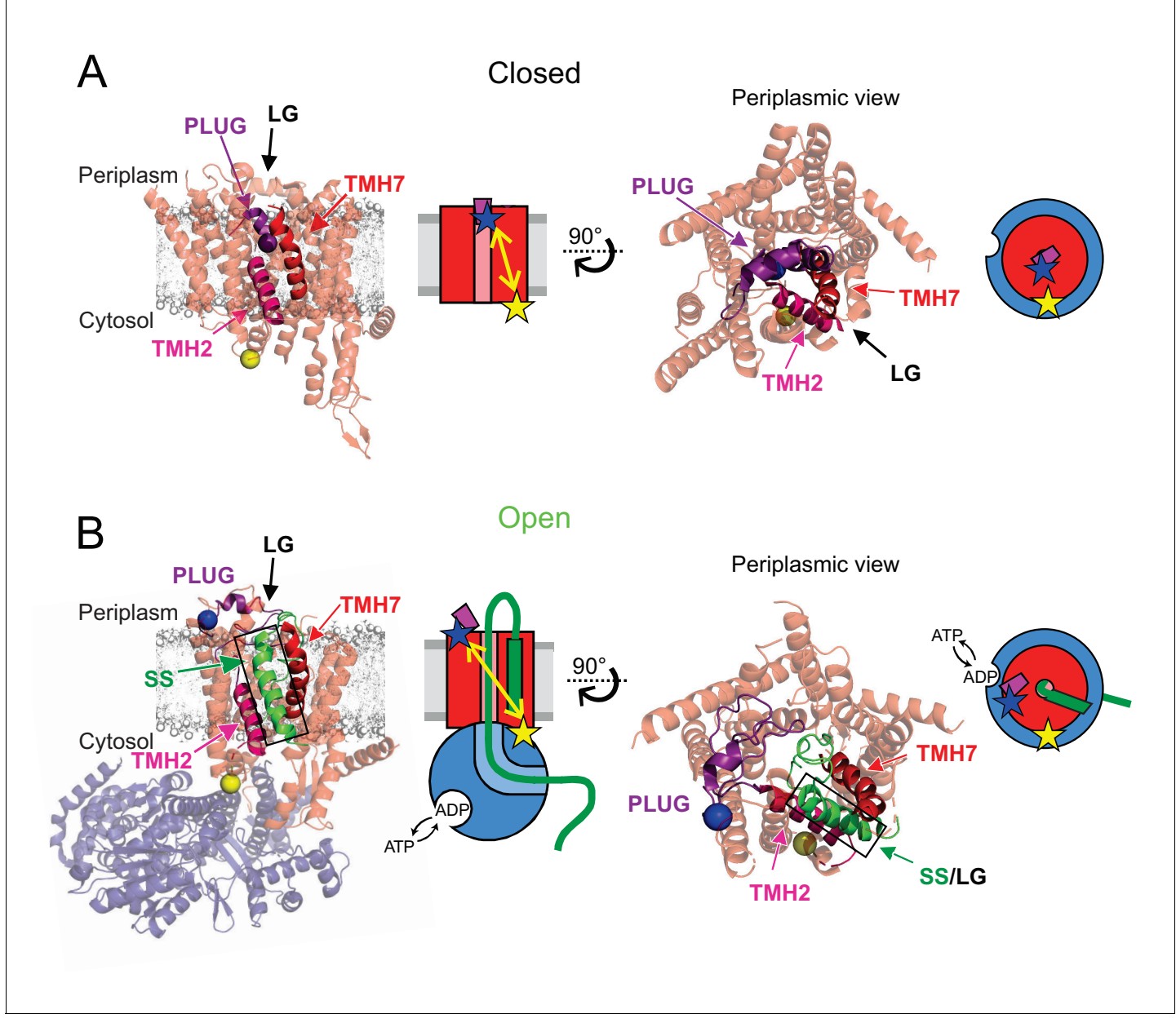

**Figure 1.** Channel opening and helical plug motion illustrated by available high-resolution structures. (**A**) Closed SecYEG (PDB: 5AWW, *Tanaka et al., 2015*). SecYEG (light red) is shown embedded in a modelled membrane (grey) with the plug helix highlighted (purple). Also identified are the transmembrane helices TMH2 (magenta) and TMH7 (red) that are demarking the lateral gate (LG). The structurally equivalent position of *E. coli* SecY residue M63 within the plug is depicted as a blue ball while a cytoplasmic side reference residue K106 is shown as a yellow ball. A side view (left) and a periplasmic view (right) are shown together with a schematic of each state (SecYEG in red, the plug in purple). The respective distances are shown as yellow double-arrow line with the dyes shown as stars (blue and yellow). (**B**) The open state SecYEG:SecA (PDB: 5EUL, (*Li et al., 2016*). Colours and labelling are as in panel A, with Sec A in transparent blue and the translocated polypeptide (green) and signal sequence (SS, green with black outline). Sec A is depicted blue in the schematics.

DOI: https://doi.org/10.7554/eLife.35112.002

The following figure supplements are available for figure 1:

**Figure supplement 1.** Modelling of fluorescent dye accessible volumes.

DOI: https://doi.org/10.7554/eLife.35112.003

**Figure supplement 2.** Activity of dual labelled SecY$_{MK}$EG in a translocation assay.

DOI: https://doi.org/10.7554/eLife.35112.004

(*Figure 1A*) (*Saparov et al., 2007*). Separation of these domains opens a channel across the membrane (secretion) as well as a lateral gate (LG) for SS docking and membrane protein insertion (*Figure 1B*). Activation is achieved by the ribosome nascent chain complex (*Jomaa et al., 2016*; *Voorhees and Hegde, 2016*), or by association of pre-protein and SecA (*Corey et al., 2016b*; *Lill et al., 1989*).

Despite numerous protein structures of the bacterial, archaeal and eukaryotic Sec systems (see (*Collinson et al., 2015*) and references therein), no consensus has yet emerged for the dynamic mechanism underlying translocation. Certainly, the bacterial initiation complex: pre-protein-SecA-SecYEG, undergoes a series of conformational changes prior to translocation. This involves the intercalation of the SS into the LG of SecY – at the interface with the lipid bilayer, between TMHs 2 and 7 (*Figure 1B*) (*Briggs et al., 1986*; *Hizlan et al., 2012*; *Li et al., 2016*; *McKnight et al., 1991*). This in turn causes TMH7 to relocate and the plug to become displaced from the channel, in a process termed 'unlocking' (*Corey et al., 2016b*; *Hizlan et al., 2012*). Meanwhile, the association of SecA causes a partial opening of the channel and the mobilisation of the pre-protein cross-linking domain of SecA, which forms a clamp around the translocating pre-protein (*Zimmer et al., 2008*). Together, these steps lead to full activation of the SecA ATPase and prime the channel to translocate the remainder of the pre-protein. However, the order of events, energy requirements and kinetics of these steps have yet to be resolved.

There is less agreement in the field as to the processive nature of translocation per se. Two principal models have been proposed: (1) a processive power-stroke in which a fixed length of substrate is transported for each ATP molecule hydrolysed by SecA (*Erlandson et al., 2008*; *Zimmer et al., 2008*); and (2) a Brownian ratchet model whereby passive diffusive motion is conferred directionality by gating at the expense of ATP hydrolysis (*Allen et al., 2016*; *Liang et al., 2009a*). Recent studies favour at least some element of diffusion: previously, we proposed a 'pure' ratchet model, in which the free energy available from ATP binding and hydrolysis at SecA drives a Brownian ratchet at the SecY LG (*Allen et al., 2016*), while others have suggested a hybrid power stroke/diffusion model, in which ATP hydrolysis generates a power stroke ('push') on the polypeptide, followed by diffusion through the pore ('slide') (*Bauer et al., 2014*). This marks a shift from purely power stroke models based on static structural snapshots to a stochastic view in which intrinsic dynamics of the complex are taken into account (*Corey et al., 2016a*).

A central problem in addressing these questions is the challenge of dissecting the rates of the various steps, and their dependence on extrinsic factors such as ATP concentration or pre-protein sequence. For example, quantification of the protein translocation rate has been undermined by problems inherent to ensemble analysis of unsynchronised reactions; wherein multiple steps are convoluted into one overall measurable value; that is an overall translocation rate or average rate of ATP hydrolysis (*Brundage et al., 1990*; *De Keyzer et al., 2002*). Furthermore, many of these 'translocation rates' are estimated from only few discrete time points that are defined by the time of protease addition to degrade untranslocated substrate, which does not instantaneously quench the reaction. This situation may explain why wildly different figures have been published for the energetic cost of transport: one study proposed a single ATP to drive the passage of ~5 kDa of protein (roughly 40 amino acids) across the membrane (*van der Wolk et al., 1997*), while a later analysis arrived at 5 molecules of ATP hydrolysed per single amino acid transported (*Tomkiewicz et al., 2006*).

Here, we exploit single molecule Förster Resonance Energy Transfer (FRET) analyses – building on a previous study (*Allen et al., 2016*) – to dissect the mechanism of protein translocation in unprecedented detail. This approach, which utilises an array of single molecule FRET experiments sensitive to different timescales, alongside ensemble measurements, allowed us to delineate several stages of translocation: (1) SS-dependent but ATP-independent unlocking of the translocon; (2) ATP-dependent plug opening; (3) a pre-processive translocation stage; (4) ATP-dependent processive translocation and (5) ATP-independent, fast channel closing. This has enabled us to estimate an intrinsic, processive translocation rate of ~40 amino acids per second. The broad distribution of the processive translocation rates is consistent with the stochastic models (*Allen et al., 2016*); (*Bauer et al., 2014*); (*Liang et al., 2009b*).

## Results

### Selection of surface residues for dye attachment

Building on the successful application of single molecule FRET to follow SecYEG opening (*Allen et al., 2016*), we utilised a similar approach to follow another key event associated with protein transport: the movement of the SecY plug during complex activation and channel opening. The plug helix is expected to relocate during activation of the channel by association of the SS and SecA, and remains open during the protein translocation process (*Bieker et al., 1990*; *Flower et al., 1995*; *Hizlan et al., 2012*; *Li et al., 2016*; *Robson et al., 2009a*; *Tam et al., 2005*; *Zimmer et al., 2008*). Met63 (*E. coli* numbering) of the plug region of SecY was selected for dye attachment in order to monitor its mobility (*Figure 1*). As FRET is most sensitive for inter-dye distances close to the Förster radius (6 nm for Alexa Fluor 488 and 594 dye pair used here), we chose the solvent-accessible residue Lys106, within the loop on the cytoplasmic side of SecY, as the reference dye attachment site (*Figure 1*).

In order to verify the suitability of this labelling scheme, the positions accessible to the attached dyes (accessible volumes) were modelled onto the available closed and open state crystal structures of SecYEG and SecA, using a Monte Carlo protocol that checks for steric clashes (*Figure 1—figure supplement 1A–D*). The resulting inter-dye distance distributions (*Figure 1—figure supplement 1E*) yielded theoretical FRET efficiency ($E_{FRET}$) histograms (*Figure 1—figure supplement 1F*) centred at low $E_{FRET}$ values (0.16–0.2) for the two open configurations sampled here and at 0.4 with ~0.6 shoulder for the closed state. These differences are sufficient to be distinguishable by single molecule FRET; hence, M63 and K106 were mutated to cysteine in a Cys-free SecYEG variant (*Deville et al., 2011*), and the resulting protein was labelled with Alexa Fluor 488 and 594 maleimide dyes (the doubly labelled protein is hereafter designated SecY$_{MK}$EG). Bulk transport assays utilising a model pre-protein substrate (233 amino acid long variant of proOmpA; see Materials and methods), showed that the labelled SecY$_{MK}$EG was fully active for transport (*Figure 1—figure supplement 2*).

### Single molecule monitoring of plug relocation

The expected $E_{FRET}$ signal during translocation is schematically depicted in *Figure 2A*. The duration (dwell time) of the open, low-FRET state is related to the duration of translocation event and thus is expected to be inversely dependent on the translocation rate, and to increase with the length of the substrate. Based on previous ensemble translocation rate estimates (*Brundage et al., 1990*; *De Keyzer et al., 2002*), the open state is expected to persist from seconds to minutes, while the rates of the transitions between the open and closed states (*Figure 2A*, red and blue dashed vertical lines) although unknown, are likely to be much faster. In order to capture the slow dwell times and potentially fast transitions, two complementary single molecule detection techniques were employed: confocal microscopy for detection of events on the millisecond timescale (*Figure 2B*) and total internal reflection (TIRF) imaging of immobilised vesicles for longer observations lasting up to several minutes (*Figure 2C*). The former allowed us to explore the rate of interconversion between closed and open states, whereas the latter was used to measure the duration of translocation and determine the translocation rate for individual SecY$_{MK}$EG complexes.

For single molecule analysis, SecY$_{MK}$EG was reconstituted into 100 nm diameter proteoliposomes formed from *E. coli* polar lipids, at concentrations to give at most one molecule of translocon per vesicle (*Allen et al., 2016*; *Deville et al., 2011*). The SecY$_{MK}$EG proteoliposomes were then either diluted to pM concentration for confocal detection (*Figure 2B*) or immobilised onto a glass surface via a biotin:streptavidin linker and imaged using TIRF microscopy (*Figure 2C*). For each setup, FRET datasets were collected in the presence of all components required for translocation: precursor protein proOmpA, the ATPase SecA, the chaperone SecB and ATP.

With confocal detection, the time-dependent fluorescent intensity traces consist of a series of short signal bursts (*Figure 2D*) that correspond to the passage of individual vesicles through the confocal volume. The duration of each burst is limited by diffusion to a few milliseconds. Bursts were extracted from the signal traces, converted to FRET efficiencies (see Materials and methods) and collated in a histogram (*Figure 2E*). The result exhibits bimodal distribution of the expected FRET efficiencies – $E_{FRET}$ ~0.2 and~0.4.

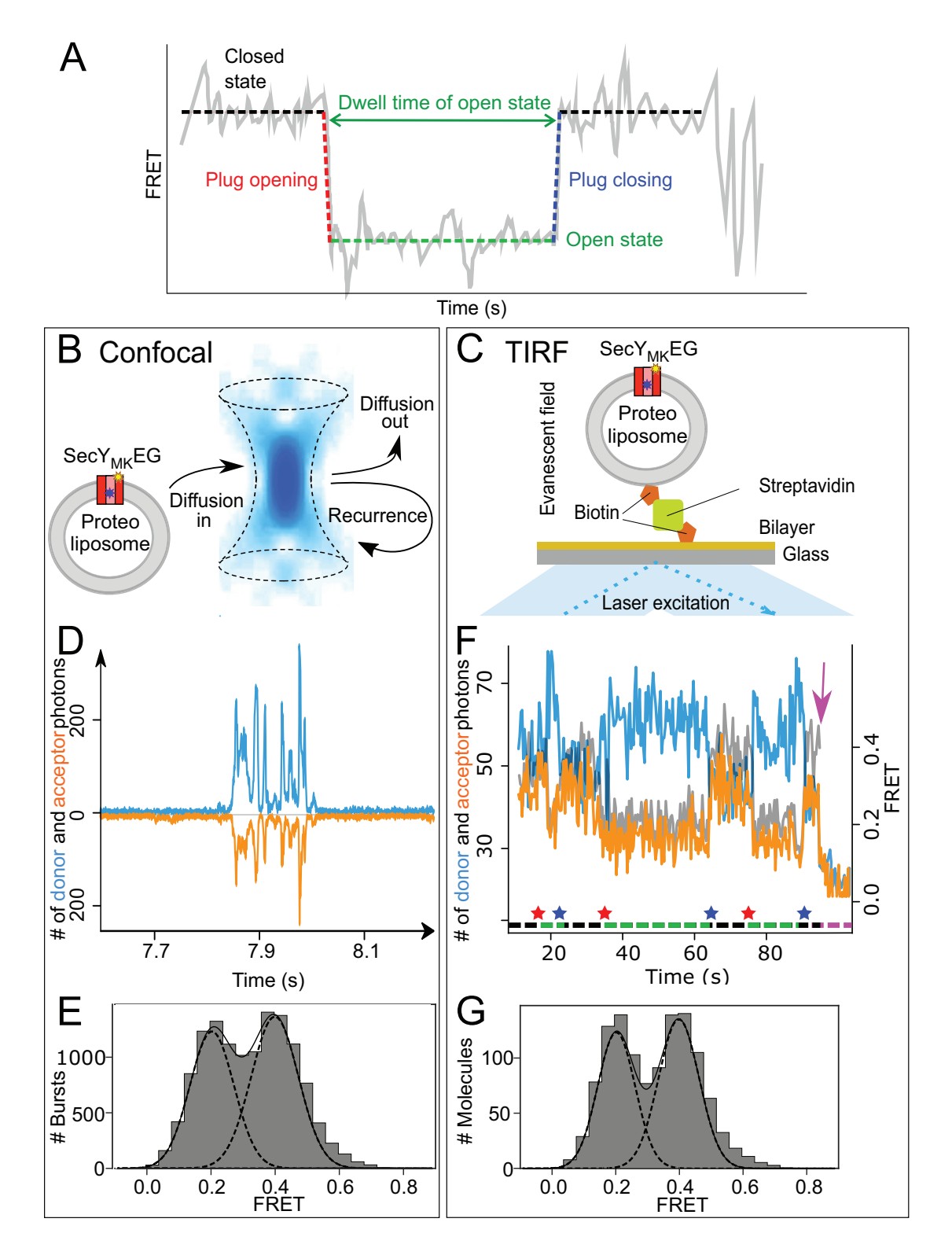

**Figure 2.** Monitoring plug movement by single molecule FRET. (**A**) Expected changes in the FRET efficiency as a consequence of plug displacement during translocation. Pre-translocation, high FRET closed state (black dashed line) changes rapidly to a low FRET, open state (red dashed line) and remains open (green dashed line) until closing (blue dashed line). (**B**) Schematic depiction of confocal (blue confocal volume) detection of freely diffusing proteoliposomes containing SecY$_{MK}$EG (red) embedded in the bilayer (grey) with recurrence and diffusion paths shown as arrows. (**C**)

*Figure 2 continued on next page*

*Figure 2 continued*

Schematic depiction of proteoliposome immobilized via a biotinylated lipid to a streptavidin (green) coated cover slip. Laser beam (blue) in a total internal reflection fluorescence (TIRF) mode creates a thin layer (~500 nm) of evanescent optical field close to the surface. (D) Example of fluorescence time traces collected in confocal microscope (donor channel -blue, acceptor channel – orange, shown with opposite sign for clarity) containing a train of bursts from recurrence. FRET data sets were collected under steady state translocation conditions, that is in the presence of short proOmpA substrate (100 aa, 700 nM), the ATPase SecA (1 µM), the chaperone SecB (10 µM) and 2 mM ATP. (E) FRET efficiency histograms derived from confocal data (10,000 events) under steady state translocation conditions. A sum (solid black line) of two Gaussian functions (black dashed lines) approximates the experimental histograms. The histogram was corrected for contribution from the 50% SecY$_{MK}$EG in opposite orientation which is unable to bind SecA and translocate (see Materials and methods and *Figure 2—figure supplement 2* for further details). (F) Example of TIRF fluorescence trace for translocation of proOmpA 100 aa substrate with dwell times on the order of seconds (donor channel is blue, acceptor orange and FRET efficiency shown in grey). The system starts in a closed state, undergoes initiation and opening of the plug (indicated by a red star below the trace), which remains open during translocation (green dashed line under the trace). After translocation is finished, the plug snaps back (blue star) to seal the pore and the system remains in the closed state (black dashed line) until another round of translocation or one of the dyes photobleaches (magenta arrow). Note that duration of the translocation events varies and reflects the stochastic nature of the process. (G) TIRF data histogram (300 events) collected during translocation of proOmpA 100 aa under steady state, multiple turnover conditions as seen in panel F above. Fitting to two Gaussians is depicted as in the panel E.

DOI: https://doi.org/10.7554/eLife.35112.005

The following figure supplements are available for figure 2:

**Figure supplement 1.** 1D FRET efficiency histograms for controls.
DOI: https://doi.org/10.7554/eLife.35112.006

**Figure supplement 2.** Correction of 1D FRET efficiency histograms for unresponsive population of SecY$_{MK}$EG in opposite orientation.
DOI: https://doi.org/10.7554/eLife.35112.007

**Figure supplement 3.** RASP workflow describing how to obtain two-dimensional FRET efficiency histograms and transition density plots from fluorescence bursts.
DOI: https://doi.org/10.7554/eLife.35112.008

The TIRF signal (*Figure 2F*) allows observation of events lasting seconds or minutes, with durations ultimately limited only by photobleaching. A photobleaching event results in a single-step, abrupt change: donor photobleaching reduces both signals to background levels (seen after approximately 90 s in *Figure 2F*, magenta arrow), while acceptor photobleaching changes the acceptor signal to background and the donor signal to maximum. Single-step photobleaching confirms that the trace corresponds to a proteoliposome containing a single copy of SecY$_{MK}$EG labelled with one copy of each dye. FRET efficiencies are computed for each time point (up to the photobleaching event) for many traces and collated into a histogram (*Figure 2G*), which again exhibits a bimodal distribution with E$_{FRET}$ ~0.2 and ~0.4.

Both methods thus yield histograms that show co-existence of two states with E$_{FRET}$ ~0.2 and ~0.4–0.6, which in turn are comparable to the values predicted for the respective open and closed states (*Figure 1—figure supplement 1F*). The similarity of the histograms obtained by the two methods also suggests no bias in the representation of each state under steady-state conditions, irrespective of whether SecY$_{MK}$EG is immobilised to a surface or placed into a freely diffusing proteoliposome in solution. The approximately equal populations of the open and closed states are a result of the steady state conditions in which complexes spend roughly an equal time unoccupied, waiting for the next initiation event (closed), or are engaged in translocation (open). As seen in the example TIRF trace in *Figure 2F*, the same complex undergoes multiple turnovers; yet, the experiments were performed in the absence of signal peptidase, required in vivo to liberate the secreted polypeptide from the plasma membrane. Therefore, the SS must be able to diffuse spontaneously away from the translocon and into the membrane. This is compatible with pre-protein maturation occurring late or post-translocation (*Josefsson and Randall, 1981*).

To confirm the assignment of the FRET populations to functional states, additional control experiments were performed using confocal microscopy, including: SecY$_{MK}$EG alone, which was expected to be closed, was examined. Surprisingly, a peak was found in the FRET histogram at E$_{FRET}$ ~ 0.35 (*Figure 2—figure supplement 1A*); that is lower than expected for the closed state observed in the structures. However, the previous steady-state translocation reactions, showing roughly equal populations of open and closed states (*Figure 2*), were performed in the presence of proOmpA, ATP and excess SecA. Hence, another control, containing SecY$_{MK}$EG, saturating SecA and ATP, but without proOmpA, was examined and indeed the FRET efficiency histogram reproduces that of the low

FRET closed state (*Figure 2—figure supplement 1B*). This suggests that the fully closed state is only attained within the SecY$_{MK}$EG:SecA complex, while SecY$_{MK}$EG alone is either undergoing dynamic rapid exchange between open and closed states, and/or samples partially open states. Further analysis (see section on translocon unlocking below) suggests the latter is the most plausible explanation.

The open state control was prepared by first saturating SecY$_{MK}$EG proteoliposomes with proOmpA and SecA in the presence of saturating ATP (1 mM), then rapidly quenching the translocation reaction by addition of excess AMP-PNP (5 mM) – a non-hydrolysable analogue of ATP, previously shown to keep the translocation complex intact (*Deville et al., 2011*). This condition resulted in a FRET distribution with a peak below 0.2 (*Figure 2—figure supplement 1C*), as expected for a trapped open state.

The distinct, intermediate nature of the SecYEG FRET efficiency histogram (*Figure 2—figure supplement 1A*) as compared to either the closed (*Figure 2—figure supplement 1B*) or the open (*Figure 2—figure supplement 1C*) state enabled the correction for the contribution of the 50% complexes embedded into the liposomes with their cytoplasmic face facing inwards; that is not able to respond to the addition of SecA and other components added to the outside of the vesicles. The corrected histograms can be readily decomposed into closed and open state contributions (*Figure 2—figure supplement 2*).

## Fast (ms) plug opening requires ATP hydrolysis

Both TIRF and confocal microscopy revealed that SecY$_{MK}$EG occupies discrete closed (inactive) and open (translocating) states under steady-state conditions. While the transitions between them are seen as instantaneous within the time resolution of the TIRF method (limited by signal to 0.2 s per frame), it is possible to resolve these events with confocal data collection on the millisecond timescale. On average bursts last for only a few milliseconds (*Figure 3—figure supplement 1A*), but it is possible to take advantage of the known behaviour of single particles in dilute solutions. As illustrated in *Figure 2B*, highly diluted diffusing particles are likely to revisit the confocal volume within a short time, while entry of another particle in the same time frame is statistically less probable. This produces a train of closely spaced bursts (*Figure 2D*), which result from multiple passages of the same vesicle through the confocal volume – a phenomenon called recurrence (*Figure 2B*) (*Hoffmann et al., 2011*). When the diffusion coefficient of the vesicles, the size and shape of the confocal volume and concentration are taken into account, the probability of recurrence of the same proteoliposome can be estimated (*Figure 3—figure supplement 1B*). In our experiments, this likelihood remained high (p>0.9) for up to 80 ms, allowing us to extend the observation window. Therefore, the experimental setup provides a large window in which bursts are likely to be generated by the same labelled SecY$_{MK}$EG as it enters and re-enters the confocal volume.

We exploited this phenomenon to follow transitions during translocation initiation with the help of Recurrence Analysis of Single Particles (RASP, [*Hoffmann et al., 2011*]), which is schematically depicted and described in *Figure 2—figure supplement 3*. In essence, this analysis results in a two-dimensional histogram (*Figure 2—figure supplement 3* bottom) which represents frequencies of transitions between E1 and E2 states with the given time window. Any off-diagonal densities (i.e. E1 $\neq$ E2) within such plot indicate that a change of state is taking place within the set time window (set at 50 ms in *Figure 2—figure supplement 3*). This approach was applied to the translocation initiation as described below.

When examining early stages in translocation, we maximised the number of initiation events by starting translocation reactions in sub-saturating ATP (0.1 mM) concentrations and measuring immediately in a pre-steady state data collection (*Figure 3A*). Complexes that remain in the same state appear as spots along the diagonal within the 2D FRET efficiency plots (*e.g. Figure 3A*, top panel), while spots off the diagonal (*Figure 3A*, middle and bottom) represent state transitions. As the time between E1 and E2 increases (indicated in panels of *Figure 3A*), the probability of a state change increases. This is illustrated in *Figure 3A*: after 1.6 ms (top panel), most of the bursts both start and end with the plug closed (E$_{FRET}$ ~ 0.4), while as time increases (middle and bottom panels) progressively more complexes transition from closed to open (i.e. appear below the diagonal, see *Figure 3—figure supplement 2* and *Video 1* for full time-resolved details). The population of states along the E2 axis between the two main spots may be a result of averaging (burst duration being on the same timescale as the transition kinetics) or, alternatively, represent *bona fide* intermediates.

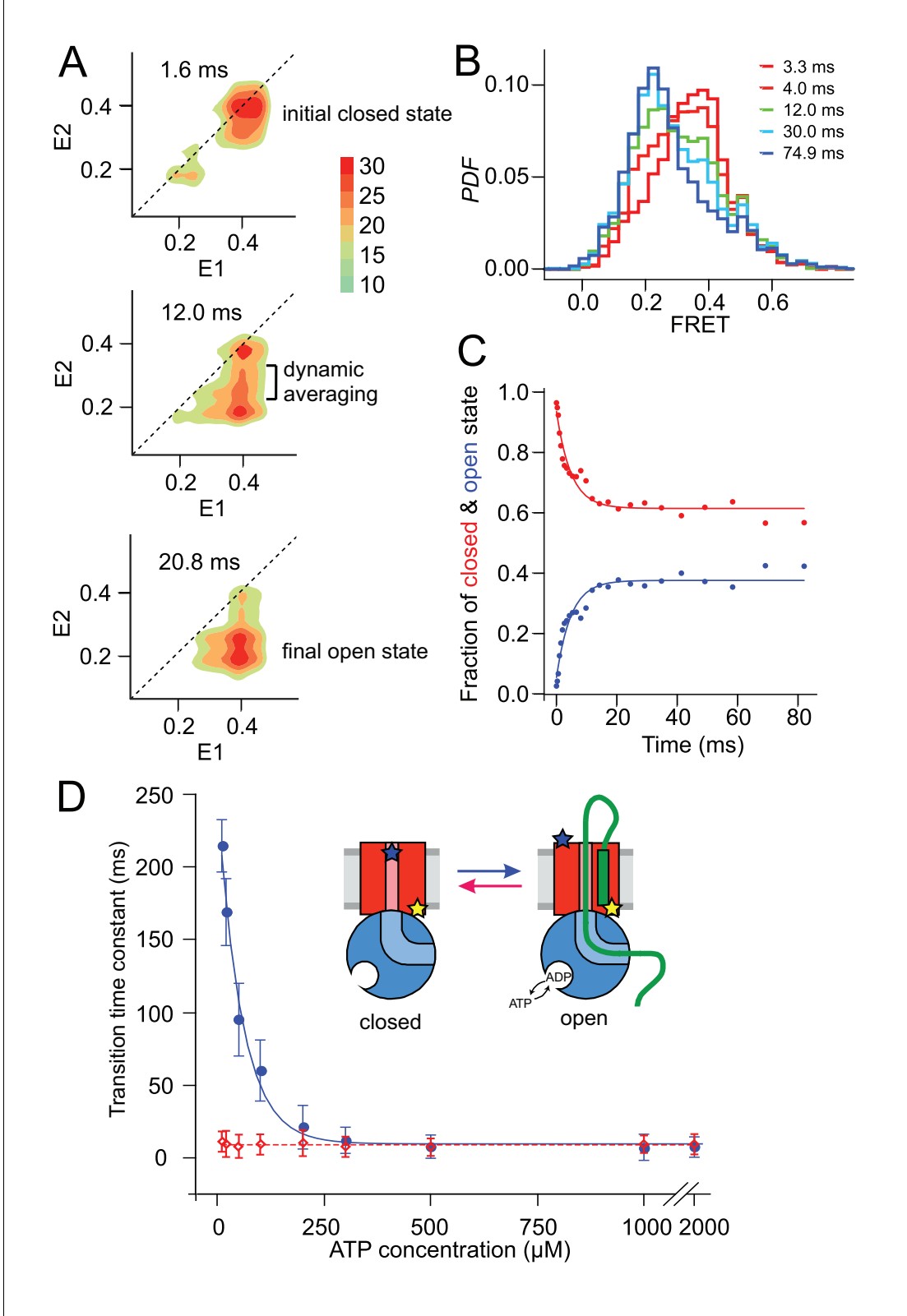

**Figure 3.** Monitoring fast plug movement during initiation. (**A**) Two-dimensional FRET efficiency contour plots (transition density plots) were obtained from bursts collected for SecY$_{MK}$EG:SecA:SecB:pOA (proOmpA 100 aa) in the presence of 0.1 mM ATP using the confocal setup. The events were classified according to initial FRET (E1) and the burst recurrence FRET (E2) observed after the indicated time delay. Data shown as Probability Density Function (PDF) contour plots with scale on the right. (**B**) RASP analysis was performed with the initial state interval of 3.9 to 4.1. The RASP PDF shows a

*Figure 3 continued on next page*

*Figure 3 continued*

rapid decreasing closed state population (red) and concomitantly increasing open state population (blue). Time in these RASP histograms is colour coded according to the legend within the panel. (**C**) Opening (blue) and closing (red) kinetic profiles extracted by a two-state approximation to the data in B). Solid lines represent least square exponential fitting of the data. (**D**) ATP concentration dependence of the opening (blue) and closing (red) transition time constants. Open and closed state interconversion is shown schematically in the centre.

DOI: https://doi.org/10.7554/eLife.35112.009

The following figure supplements are available for figure 3:

**Figure supplement 1.** Burst duration and recurrence probability.
DOI: https://doi.org/10.7554/eLife.35112.010

**Figure supplement 2.** Two-dimensional transition density plots.
DOI: https://doi.org/10.7554/eLife.35112.011

**Figure supplement 3.** Spontaneous opening and closing in the absence of translocation substrate.
DOI: https://doi.org/10.7554/eLife.35112.012

**Figure supplement 4.** RASP derived two-dimensional transition density plots in the presence of AMP-PNP.
DOI: https://doi.org/10.7554/eLife.35112.013

**Figure supplement 5.** Activation energies for plug opening and closing.
DOI: https://doi.org/10.7554/eLife.35112.014

The RASP approach effectively allows kinetic rate constants to be determined without the need to synchronise the sample at the single molecule level. As long as the experimental conditions assure that the transitions of interest are taking place, interconversion between different states can be quantified by collating information from many single molecule events. One way to do this is to select initial bursts with a specific E1 value, and follow the time evolution of E2. For example, to monitor plug opening, we selected bursts with starting $E_{FRET}$ values within a window of 0.3–0.5 (closed; centred at the E1 ~0.4) then plotted E2 after different times in a one-dimensional histogram (**Figure 3B**). At each time point, the histogram is well described by a two-state model, allowing the open and closed populations to be determined. Fitting these populations as function of time yields the rate of plug opening (**Figure 3C**). The rate of plug closing can be determined in the same way; both occur on a millisecond timescale.

The dependence of these transition times on the concentration of ATP (or of non-hydrolysable ATP analogues) was next investigated to reveal whether plug opening or closing depend on ATP binding and/or hydrolysis. **Figure 3D** shows the opening time constant in the presence of SecA, SecB and proOmpA increases with decreasing ATP concentration, while closing happens on a ~ 10 ms timescale and does not depend on ATP. The opening time constant also converges to 10 ms in a saturating ATP concentration. The apparent concentration at which the opening process is at its half maximum rate, $K_{50\%}$ ~ 56 µM, obtained by fitting these data, is similar to that of the SecA ATPase $K_M$ ~50 µM (**Robson et al., 2009a**), suggesting that ATP hydrolysis is required for the initial plug opening. In the absence of proOmpA there are only rare spontaneous opening and closing events, with time constants that do not depend on ATP concentration (**Figure 3—figure supplement 3**). In the presence of SecA, proOmpA and AMP-PNP, $SecY_{MK}EG$ remains in the closed state (**Figure 3—figure supplement 4**). Therefore, ATP binding and hydrolysis together with pre-protein substrate engagement are needed to fully displace the plug from the channel.

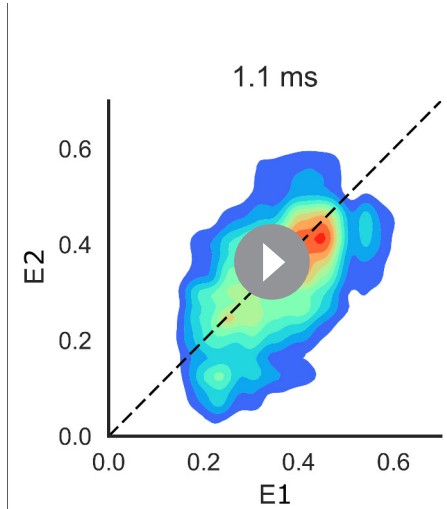

**Video 1.** Detailed sampling of time resolved transition density plots shown in **Figure 3A**. Conditions: $SecY_{MK}EG$:SecA:SecB:pOA in the presence of 0.1 mM ATP.
DOI: https://doi.org/10.7554/eLife.35112.015

We also examined temperature dependence of opening and closure, over the range 15–37°C (288–310 K) for which the membrane remains fluid and SecY$_{MK}$EG is active (*Figure 3—figure supplement 5*). The activation energy for opening, E$_a$ ~61 kJ/mol, is close to the value measured for SecA ATPase activity in the presence of translocating substrate or signal peptide (66 kJ/mol), but much lower than that of SecY$_{MK}$EG:SecA alone (~180 kJ/mol) (*Gouridis et al., 2009*). This suggests that the ATP-driven plug opening is performed by SecA in complex with SecY$_{MK}$EG pre-activated by the SS. Furthermore, the plug closing exhibits significantly lower activation energy – E$_a$ ~45 kJ/mol, which is below values reported for ATPases (60–70 kJ/mol) (*Jenkins et al., 1999*), consistent with the results shown in *Figure 3D* that closing is independent of ATP.

## Translocon unlocking by the signal sequence is necessary, but not sufficient, for plug opening

Next, we used RASP analysis, as described above, to further probe the determinants of plug motion. SecY$_{MK}$EG alone shows considerable static heterogeneity (spread along the diagonal) and dynamics (off-diagonal spots), with a broad diffuse spot between the closed and open configurations (*Figure 4A*). However, judging from the absence of density at the stationary position of the open state (E1 = 0.2, E2 = 0.2 area in *Figure 4A*, area marked with *) the open state is only transiently populated and returns to a partly open configuration (E1 = 0.2, E2 = 0.3 density region in *Figure 4A*, area marked with #). The transient nature of these conformational diversions is further demonstrated by only a small contribution of the open state to the equilibrium histogram in *Figure 2—figure supplement 1* (<7%) and is consistent with low ion conductivity of SecYEG (*Saparov et al., 2007*). By contrast, the plug populates predominantly the expected closed state (E$_{FRET}$ ~0.4) in a plug stabilising mutant SecY$_{R357E}$ (the labelled mutant designated SecY$_{MK,R357E}$EG) (*Figure 4—figure supplement 1*) (*Tam et al., 2005*). This finding is consistent with previous reports showing the plug can be localised in the open state by disulfide bond cross-linking with SecE (*Harris and Silhavy, 1999*). These cross-links are reduced when the R357E mutant is incorporated, due to its tendency to retain the plug in the central closed position (*Tam et al., 2005*). Thus, the FRET distributions reflecting open and closed channel states are predictably affected by variants of SecYEG, which favour the latter. This suggests the attached dyes have not drastically perturbed the properties of the channel.

Addition of SecA to SecY$_{MK}$EG results in the formation of a more stable complex, with the plug predominantly residing in the high-FRET (closed) state (*Figure 4B*). This observed state is similar to the behaviour of the SecY$_{MK,R357E}$EG variant, wherein the plug is stabilised in the central closing position, even in the absence of SecA (*Figure 4—figure supplement 1*). Hence, this conformation is most likely a result of allosteric action of SecA, rather than a simple consequence of a direct SecA interaction with the attached dyes.

The active translocon was assembled by addition of proOmpA (100 aa long variant), SecB and saturating ATP to SecY$_{MK}$EG:SecA. The RASP analysis at steady-state (>5 min) shows, as expected, that the plug shifts into a predominantly open state, with some minor closing transitions represented by a smear to higher FRET along the E1 axis (*Figure 4C*). This behaviour is distinct from the same active complex at sub-saturating ATP concentration (0.1 mM), which is much more dynamic (*Figure 4D*): under these conditions, the FRET landscape is characterised by a significant population of the closed states transitioning to the open state (*Figure 4D*, off-diagonal smear along E2 axis). This confirms that plug opening is indeed dependent on ATP.

It has been shown previously that the SS can unlock the translocon in trans and initiate the translocation of mature substrates, that is those lacking a SS (*Gouridis et al., 2009*). Structural studies demonstrated that this unlocking process involves SS intercalation at the lateral gate of SecY (LG, between TMH2 and TMH7, *Figure 1*), which can occur even in the absence of SecA (*Hizlan et al., 2012*). Furthermore, this association has also shown to displace the plug from the central location in SecY (*Hizlan et al., 2012*).

The interaction of the SS and mature regions of the pre-protein with the translocon were further explored here, with respect to plug dynamics. For these experiments, OmpA (*i.e.* proOmpA lacking a SS) and/or a synthetic peptide representative of the SS (see Materials and methods) were added to SecY$_{MK}$EG:SecA in the presence of saturating ATP for analysis of the resultant FRET landscape. The addition of OmpA and SS resulted in plug opening (*Figure 4E*) akin to that observed for

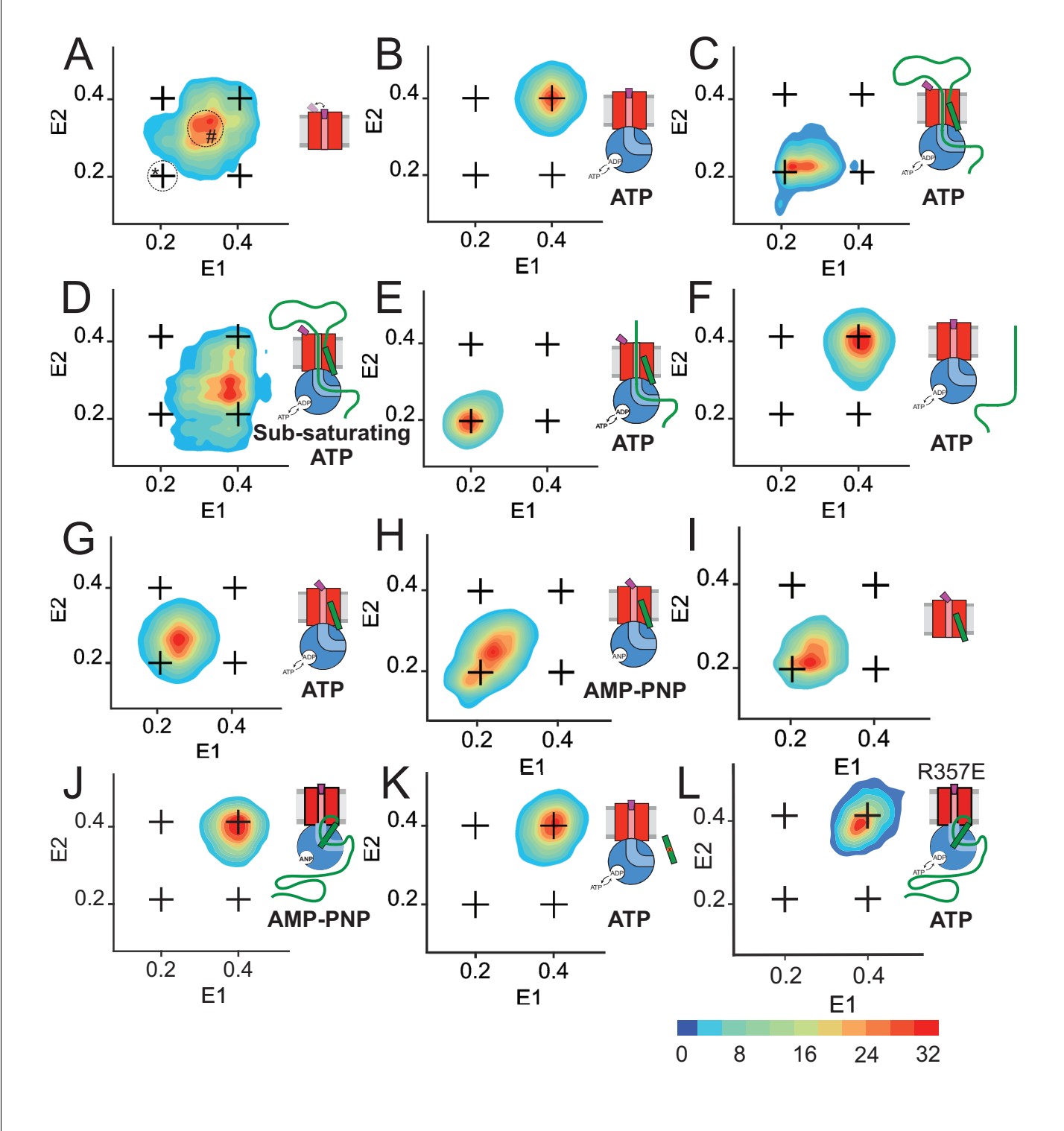

**Figure 4.** Two-dimensional FRET efficiency histograms detect transitions between states. (**A**) SecY$_{MK}$EG alone. Transition density for delays up to 21 ms were obtained from RASP analysis of 10,000 events. In all panels, the crosshair symbols indicate positions of the open and closed state FRET values within the E1-E2 plot. A scale bar for count contour levels is shown in the lower right corner. A cartoon in each panel schematically depicts the composition of the SecYEG complex and reaction conditions. In panel A, the positions of stably open state and the stationary part-open state are circled and marked with * and #, respectively. (**B**) SecY$_{MK}$EG:SecA in the presence of 1 mM ATP. (**C**) SecY$_{MK}$EG:SecA:SecB:pOA in the presence of 1 mM ATP. (**D**) SecY$_{MK}$EG:SecA:SecB:pOA in the presence of 0.1 mM ATP (sub-saturating condition). (**E**) SecY$_{MK}$EG:SecA:SecB:OmpA (lacking SS) in the presence of SS peptide added in trans and 1 mM ATP. (**F**) SecY$_{MK}$EG:SecA:SecB:OmpA in the presence of 1 mM ATP. (**G**) SecY$_{MK}$EG:SecA:SecB in the

*Figure 4 continued on next page*

Figure 4 continued

presence of SS peptide and 1 mM ATP. (H) SecY$_{MK}$EG:SecA:SecB in the presence of SS peptide and 1 mM AMP-PNP (depicted as ANP in the cartoon). (I) SecY$_{MK}$EG in the presence of SS. (J) SecY$_{MK}$EG:SecA:SecB:pOA in the presence of 1 mM AMP-PNP (ANP in the cartoon). (K) SecY$_{MK}$EG:SecA:SecB in the presence of defective (four residue deletion) SS peptide and 1 mM ATP. (L) SecY$_{MK,R357E}$EG:SecA:SecB:pOA in the presence of 1 mM ATP.

DOI: https://doi.org/10.7554/eLife.35112.016

The following figure supplements are available for figure 4:

**Figure supplement 1.** FRET characterization of SecY$_{MK, R357E}$EG 'closed' plug mutant.

DOI: https://doi.org/10.7554/eLife.35112.017

**Figure supplement 2.** Burst variance analysis for selected states.

DOI: https://doi.org/10.7554/eLife.35112.018

addition of proOmpA (*Figure 4C*). Therefore, the SS can indeed act in trans. As expected OmpA alone did not bring about plug opening (*Figure 4F*).

Interestingly, when the SS was added to SecY$_{MK}$EG:SecA in the absence of OmpA, with either ATP or AMPPNP (*Figure 4G,H*), or even with SecA also absent (*Figure 4I*) – the plug assumes a new aparently static state with E$_{FRET}$ ~ 0.25: note diagonal localisation within the E1-E2 transition density plot (*Figure 4G–I*)). This result could be a consequence of a *bona fide* new state, or dynamic averaging between two distinct states on a sub-millisecond timescale that is too rapid to resolve by RASP. In order to distinguish between these possibilities, we performed Burst Variance Analysis (BVA) of the data on a sub-millisecond (0.1 ms) timescale, which can also be applied to the analysis of dynamic FRET distributions (*Torella et al., 2011*). The BVA strategy sub-divides individual bursts into contiguous sub-bursts consisting of a fixed number of photons, which are then compared with respect to acceptor photon variance of all sub-bursts within each burst. This variance is then compared to the theoretical shot-noise-limited variance. An empirical average variance of sub-bursts larger than the shot-noise-limited predicted variance for specific FRET regions would indicate the presence of state interconversion on the given fast burst timescale. A static example is illustrated in *Figure 4—figure supplement 2A* for the SecY$_{MK}$EG:SecA complex locked in the closed state. In contrast, a dynamic complex, for example the active translocating SecY$_{MK}$EG:SecA in the presence of ATP, is shown in *Figure 4—figure supplement 2B*. BVA of the SecY$_{MK}$EG:SecA:SS complex and SecY$_{MK}$EG:SS (*Figure 4—figure supplement 2C,D*, respectively) clearly remain static on the sub-millisecond timescale (compare with *Figure 4—figure supplement 2A,B*). While we cannot rule out dynamic averaging on a faster timescale, such a scenario seems unlikely given that even the fast, ATP independent plug closure takes a few milliseconds to complete (*Figure 3D*) and this timescale is accessible through both RASP and BVA. Thus, we conclude that the E$_{FRET}$ ~ 0.25 is indeed a genuine, part open state. Such assignment is consistent with the previous observation of increased plug crosslinking to SecE upon prolonged incubation with a pre-protein, which was interpreted as partial opening of the plug (*Tam et al., 2005*).

The E$_{FRET}$ ~0.25 represents a state that is clearly distinct from the closed (0.4) and open (0.2) states and is more static than SecY$_{MK}$EG alone (*Figure 4A* and *Figure 4—figure supplements 1* and *2*). We assign this newly observed intermediate state as the 'unlocked' configuration of the translocon, presumably characterised by the structure of SecYEG bound to the SS (*Hizlan et al., 2012*). The results presented here also confirm that binding of the SS is sufficient to unlock the complex (E$_{FRET}$ ~0.25), while full opening (E$_{FRET}$ ~0.2) is only achieved in the presence of complete protein substrate and ATP hydrolysis by SecA. Indeed, in the presence of non-hydrolysable AMP-PNP SecY$_{MK}$EG:SecA:SecB:proOmpA remains fully closed (*Figure 4J*), as is the case when the SecY$_{MK}$EG:SecA:SecB complex is presented with a defective SS, wherein four critical residues of the hydrophobic core were deleted proOmpA$_{\Delta IAIA7-10}$ (*Emr et al., 1980*) (*Figure 4K*). The confirmation that the SS is alone is critical for the unlocked complex suggests that ATP hydrolysis is required for SecYEG to achieve a fully open channel and for initial intercalation of the mature region of the pre-protein.

As expected, the plug stabilising mutant (SecY$_{MK,R357E}$EG) fails to open in conditions that would normally promote translocation. The presence of SecA eliminates the part open ensemble (compare *Figure 4L* and *Figure 4—figure supplement 1A*) in a fashion similar to the wt SecY$_{MK}$EG:SecA (*Figure 4B and J*), indicating that this mutation, while located on the cytoplasmic side close to the SecA binding site, does not disrupt SecA binding. The R357E mutation was originally shown to disfavour the SecY dimer formation (*Tam et al., 2005*). However, under single molecule conditions

(extremely low concentrations), the formation of such a dimer is unlikely. Thus, the R357E substitution likely disrupts an allosteric path within SecYEG that is involved in the coupling ATPase cycle of SecA to the plug opening.

## Translocation rate and slow post-initiation stage

If the low FRET state observed in the TIRF FRET traces ($E_{FRET}$ ~0.2, *Figure 2F*) represents SecY$_{MK}$EG plug opening while the substrate is being translocated, then the duration of this state should increase with the length of the pre-protein substrate. Therefore, various length proOmpAs were prepared ranging from 100 to 683 aa (where full length proOmpA itself is 354 aa; *Figure 5A*). All these substrates were transported successfully into proteoliposomes as shown by ensemble translocation assays (*Figure 5—figure supplement 1*) and by the stimulation of the ATPase activity of SecA (*Figure 5—figure supplement 2*).

TIRF traces were obtained for the proOmpA length variants, and the dwell times of the open state for each were extracted and collated into histograms (*Figure 5A* and *Figure 5—figure supplement 3*). As expected for a stochastic process, a broad distribution of dwell times was obtained for each substrate length. However, the dwell times of the open state clearly increases with the length of the substrate (*Figure 5A*). By contrast, a similar analysis for the closed state ($E_{FRET}$ ~0.4) shows no dependence of the dwell time on the length of polypeptide chain (*Figure 5—figure supplement 4*). This is not surprising: the closed state dwell time represents an average wait prior to, or between, translocation events and thus reflects the rate of assembly of the active complexes. The formation of this activated translocation complex would be expected to depend on the concentrations of individual components and other reaction conditions, which were kept constant throughout the experiments.

If the dwell time of the open state is indeed measuring translocation time, then it should increase at sub-saturating ATP concentrations. As shown in *Figure 5B*, this is indeed the case, and the dependence is consistent with a steady state $K_M$ ~ 50 µM as measured previously for the SecA ATPase (*Robson et al., 2009b*). Together, the dependence of the dwell time on pre-protein length and the short opening and closing times (see above), justifies the use of the open state dwell times as a proxy for the duration of individual translocation events. The duration of these events can then be combined to determine the rate at which the polypeptide is being translocated.

To extract the translocation rate per residue, the dwell times were plotted as a function of substrate length and the slope was obtained using a linear fit (ordinary least squares method using all dwell times, *Figure 5A*). Assuming a linear dependence of the open state dwell time on the substrate length, the slope yields the average time necessary to translocate a single amino acid. The resulting rate is ~40 ± 6 aa/s and within statistical error is independent of the cytoplasmic chaperone SecB (*Figure 5A*).

An extrapolation of the least squares fit line to zero substrate length suggests that there is a significant fixed, length-independent dwell time component of ~5 s in the absence of SecB (*Figure 5A*). This may be related to a slow initial translocation phase after the channel is already opened. Alternatively, it could represent the time it takes for the plug to return to the closed state after the translocation event is complete. However, this constant dwell time decreases close to zero in the presence of SecB. As SecB acts on the cytoplasmic side, this suggests this slow phase is related to the delivery of the substrate to SecA – with SecB presumably making the initial handover and delivery of substrate more efficient. This is consistent with the observation that SecB reduces ATP consumption, as indicated by ensemble ATPase assays with SecA (*Figure 5—figure supplement 2*). Given the plug closure time is less than 10 ms (*Figure 3D*) and SecY$_{MK}$EG complex alone seldom visits the open state in equilibrium (*Figure 2—figure supplement 1A*) the retention of the open state after the completion of the translocation is unlikely. Hence, we assign the constant dwell time to a slow initial translocation phase, which follows ATP-driven opening of the plug and is accelerated by SecB.

## Discussion

While structural methods can provide exquisite detail of specific intermediates in a protein reaction cycle, it is often difficult to position these snapshots in sequence and timescale to create an understanding of biological mechanism. This is especially so when dealing with multi-subunit membrane protein assemblies with complex substrates like biopolymers. Likewise, it is difficult to observe and

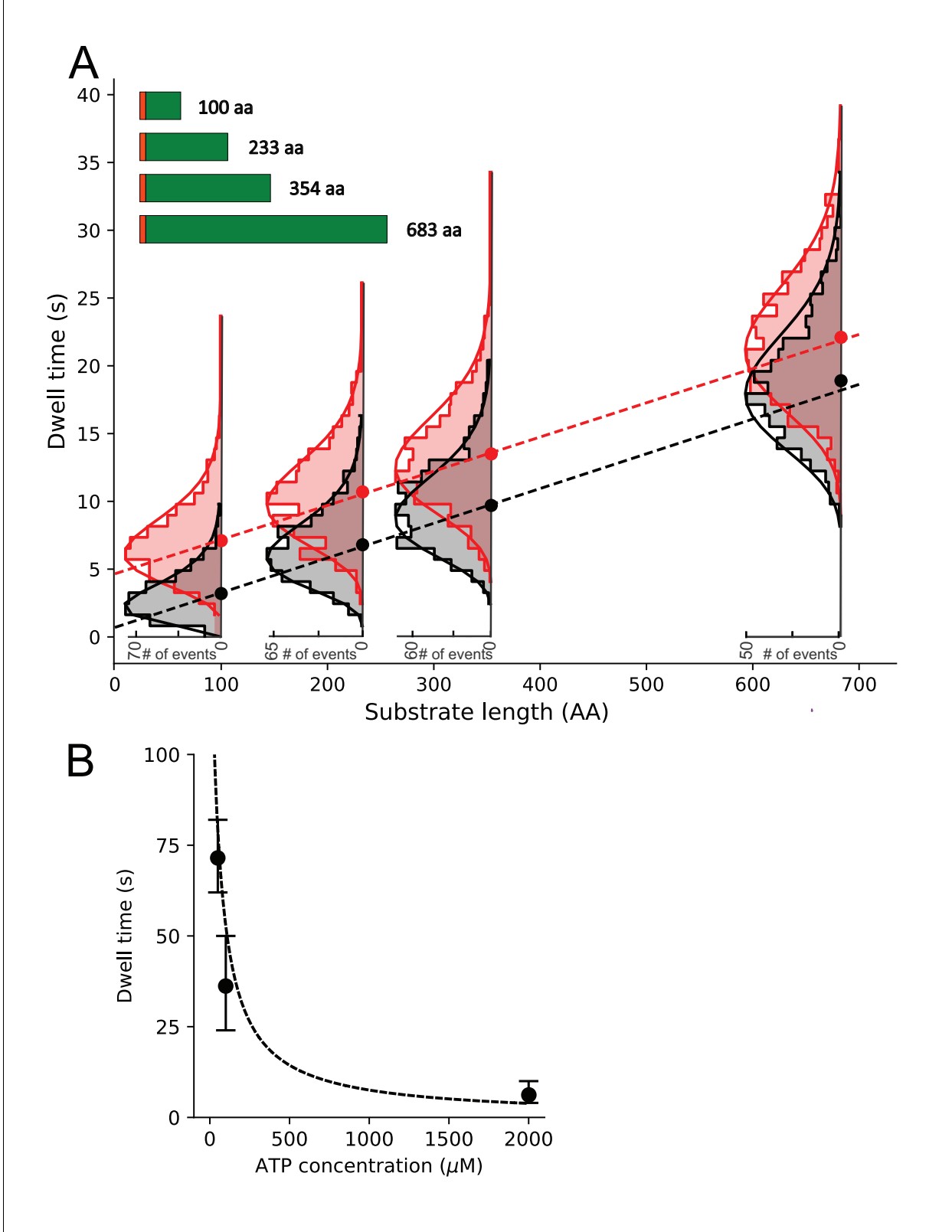

**Figure 5.** Substrate length dependence of dwell times and determination of translocation rates. (**A**) Dwell time dependence for open state ($E_{FRET}$ ~0.2) on the substrate length (schematically shown in the inset, SS depicted as orange bar) in the presence (black) or absence (red) of SecB. Ordinary least squares (OLS) linear regression (dashed lines) on all photobleaching corrected dwell times (see *Figure 5—figure supplement 3* for comparison of uncorrected and corrected dwell time distributions) gave slopes corresponding to translocation rates of 39.6 ± 6.0 aa/s (±standard error) in the absence

*Figure 5 continued on next page*

*Figure 5 continued*

of SecB and 39.0 ± 6.2 aa/s in the presence of SecB. OLS analysis of the sample with SecB resulted in an intercept close to zero (0.5 ± 0.3 s) while in the absence of SecB the intercept is approximately 5 s (4.7 ± 0.3 s). Overlaid are photobleaching corrected dwell time histograms with gamma function fits (solid lines). Only the distributions for the longest substrate were significantly affected by photobleaching (see *Figure 5—figure supplement 3* for comparison and Materials and methods for description of the deconvolution correction). (B) Average open state dwell time dependence on ATP concentration for the shortest 100 aa proOmpA substrate in the presence of SecB. Error bars were derived from the distributions of dwell times. The dashed line represents a steady state model with $K_M$ fixed at 50 μM and an amplitude scaled to the data (note that photobleaching precluded collection of more data at low ATP concentrations and thus a statistically sound fit to the data could not be performed).

DOI: https://doi.org/10.7554/eLife.35112.019

The following figure supplements are available for figure 5:

**Figure supplement 1.** Translocation of proOmpA constructs with different lengths.

DOI: https://doi.org/10.7554/eLife.35112.020

**Figure supplement 2.** Ensemble ATPase activity stimulated by proOmpA constructs with different lengths.

DOI: https://doi.org/10.7554/eLife.35112.021

**Figure supplement 3.** Deconvolution of photobleaching effect from the dwell time distributions for the longest 683 aa substrate.

DOI: https://doi.org/10.7554/eLife.35112.022

**Figure supplement 4.** Dwell time of the closed $E_{FRET} \sim 0.4$ state as a function of the translocating substrate length.

DOI: https://doi.org/10.7554/eLife.35112.023

characterise intermediates in complex reaction cycles that cannot be synchronized. Single molecule techniques are able to overcome both of these problems, providing high sensitivity, time-resolved detection of otherwise elusive intermediates, without the need for a synchronized reaction.

Here, we have combined two complementary single molecule fluorescence techniques to examine the process of pre-protein translocation through the bacterial translocon on timescales ranging from sub-milliseconds to minutes. A judiciously placed pair of reporter dyes on the SecY plug and a cytoplasmic reference site allowed us to exploit FRET changes to monitor opening and closing of the channel and to detect intermediates in the translocation process. We can now present a full cycle of events and intermediates starting from the resting state via the already known unlocked translocon (*Hizlan et al., 2012*) through a new, ATP-dependent plug opening, followed by a slow translocation phase and rapid closure (*Figure 6*).

The single molecule analysis revealed considerable dynamics of the plug within the resting translocon (state (2) in *Figure 6*); consistent with observed variations of its position within the SecYEG complex (*Van den Berg et al., 2004*; *Tanaka et al., 2015*; *Li et al., 2016*; *Zimmer et al., 2008*). RASP analysis revealed spontaneous fluctuations on the millisecond timescale as evidenced by a broad range of states accessible (*Figure 4A*), but showed a more stable closed state when bound to SecA (*Figure 4B*). This phenomenon might help retain the channel seal while the SecYEG complex is partially opened by SecA just prior to unlocking and substrate entry (state (1) in *Figure 6*) (*Zimmer et al., 2008*).

Biochemical and structural methods have described an activation, or 'unlocking', of the protein channel by the SS (*Corey et al., 2016b*; *Gouridis et al., 2009*; *Hizlan et al., 2012*). We show here that the SS alone is sufficient to partially displace the plug, but not to fully open the channel. For this, the SS and the succeeding stretches of the mature polypeptide chain need to engage the channel, in conjunction with SecA (*Tsirigotaki et al., 2018*). Crucially, we show that this state also requires ATP hydrolysis when SS is presented as part of pre-protein (states (3) and (4) in *Figure 6*).

The ATP dependence of the rate of channel opening ($K_M \sim 50$ μM) and its apparent activation energy (61 kJ/mol) mirror that of the activated SecA ATPase (*Gouridis et al., 2009*; *Robson et al., 2009b*). These combined observations suggest that the rate-limiting steps of channel opening and the hydrolytic cycle of ATP are identical. The simplest explanation is that the initial rate-limiting step is the ATP-driven intercalation of pre-protein into the channel, forcing the SS and N-terminal hairpin of the mature sequence towards the periplasm with concomitant displacement of the plug (transition (4) to (5) in *Figure 6*).

Following the fast channel opening and substrate intercalation, a slow phase (up to 5 s in the absence of SecB), independent of substrate length takes place (transition (5) to (6) in *Figure 6*). During this phase, the plug is already opened but translocation might be arrested or is very slow. This phase could be associated with a slow conformational re-arrangement of the translocon, for example

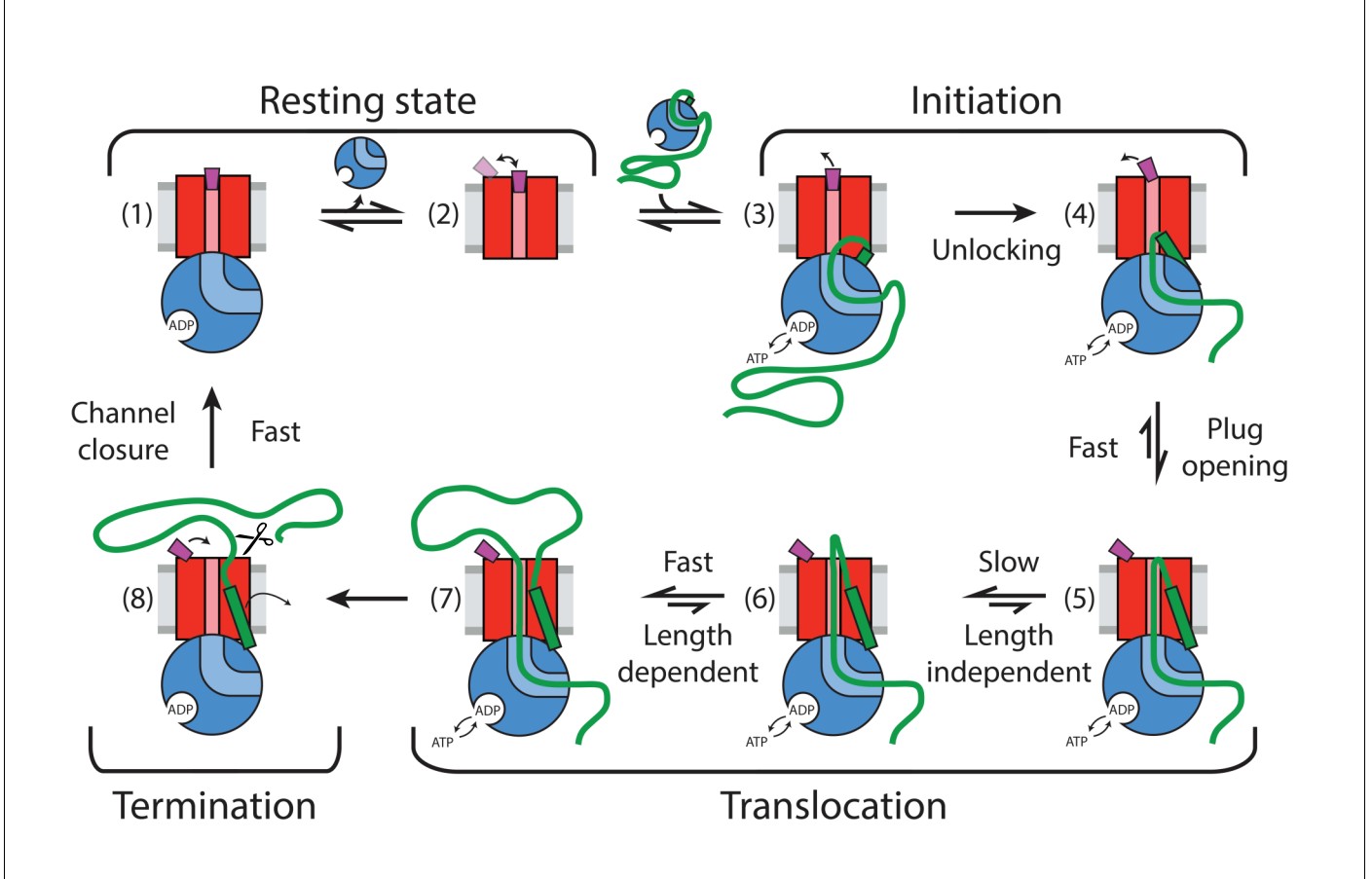

**Figure 6.** Summary of detected plug states and initiation and translocation stages. Colour coding: SecYEG – red, SecA – blue, proOmpA/OmpA – thick green line, SS –green rectangle, stationary plug – purple, plug in intermediate or transient state – magenta, lipid bilayer – grey. Scissor symbol indicates substrate liberation by signal peptidase (if present). Thin arrows indicate motion of plug and SS.
DOI: https://doi.org/10.7554/eLife.35112.024

repositioning of the SS within the LG or threading of the initial loop more completely through the translocon. Given that the duration of this phase is reduced significantly by SecB on the cytoplasmic side of the membrane, we speculate that the slow phase might be related to the unfolding of residual structure in the polypeptide substrate by the SecA:SecB complex, making the initial delivery of the substrate to SecA more efficient. In the absence of SecB, this might require multiple rounds of ATP hydrolysis with little translocation achieved. This is further supported by the higher consumption of ATP by short polypeptide substrates (*Figure 5—figure supplement 2*), the increased ATP utilization efficiency for such substrates in the presence of SecB (*Figure 5—figure supplement 2*) and the previously observed compensatory genetic link between the SecA and SecB genes (*Cook and Kumamoto, 1999*).

The slow phase is followed by a substrate chain-length-dependent phase, which occurs at an average rate of ~40 aa/s (state (6) to (7) in *Figure 6*). For each length of substrate, the distributions of rates vary considerably (*Figure 5A*), as expected for a stochastic or diffusion mediated process involving long heterogeneous substrates and is consistent with previously proposed models (*Allen et al., 2016*; *Bauer et al., 2014*). The rate of processive translocation is higher than the previously reported value of ~30 aa/s which were also estimated using proOmpA substrates with various chain lengths, but in the absence of SecB (*De Keyzer et al., 2002*; *Tomkiewicz et al., 2006*). The difference is readily explained by the slow translocation phase, which is substantial only in the absence of SecB. In the presence of SecB, the rate estimated here would support transport of typical

individual precursors (100 to 1000 amino acids) within 25 s which is close to the previously estimated transport requirements (~20 s per substrate) in rapidly dividing *E. coli* cells (*Brundage et al., 1990*).

Channel closure monitored by the relocation of the plug is fast (<10 ms) and ATP independent. Hence, it does not seem to impose any limitation on the overall translocation efficiency (state (7) to (8) in *Figure 6*). Evidently, a single translocon is capable of multiple rounds of translocation, at least for proOmpA, even in the absence of signal peptidase. This suggests that the signal peptide together with the rest of the substrate is able to diffuse from the translocon and does not perturb plug closure (state (8) to (1) in *Figure 6*).

In summary, we have devised a novel single molecule assay that allowed the determination of the intrinsic rate of polypeptide translocation for the first time. We have also characterised previously known steps in the initiation of translocation and discovered new stages in the reaction, all of which were mapped onto the overall cycle of translocation. Together the results provide a refined framework for understanding the molecular mechanism of ATP-driven protein translocation that draws on the powers of single molecule measurements to unpick complex reaction mechanisms in unsynchronised systems.

# Materials and methods

**Key resources table**

| Reagent type (species) or resource | Designation | Source or reference | Identifiers | Additional information |
|---|---|---|---|---|
| Software, algorithm | iSMS software | (*Preus et al., 2015*) - doi:10.1038/nmeth.3435 | | |
| Software, algorithm | LabView | (*Lee et al., 2005*) - doi: 10.1529/biophysj.104.054114 | | |
| Software, algorithm | FRETbursts | (*Ingargiola et al., 2016b*) - doi: 10.1371/journal.pone.0160716 | | |
| Software, algorithm | photon-hdf5 | (*Ingargiola et al., 2016a*) - doi: 10.1016/j.bpj.2015.11.013 | | |
| Software, algorithm | dual-channel burst search | (*Nir et al., 2006*) - DOI: - 10.1021/jp063483n | | |
| Software, algorithm | regularized inverse transform | (*Provencher, 1982*) - DOI10.1016/0010-4655(82)90173-4 | | |
| Software, algorithm | graphics: library Seaborn, based on Matplotlib | (*Hunter, 2007*) - DOI10.1109/MCSE.2007.55 | | |

## Protein preparation

Site-directed mutagenesis was performed using the QuikChange protocol (Agilent) and confirmed by sequencing. SecY$_{MK}$EG, SecY$_{MK,R357E}$EG, SecA, SecB and full length proOmpA were produced as described previously (*Deville et al., 2011*; *Gold et al., 2007*; *Whitehouse et al., 2012*). Different proOmpA lengths were produced adopting existing methods (*De Keyzer et al., 2002*). OmpA lacking the SS was purified as described in *Schiffrin et al. (2016)*. SecY$_{MK}$EG was produced in the same way as wild-type, then labelled for 45 mins on ice at 50 µM with 100 µM each of Alexa 488-C$_5$-maleimide and Alexa 594-C$_5$-maleimide (Invitrogen). The reactions were quenched with 10 mM DTT, and excess dye removed by gel filtration (Superdex-200, GE Healthcare, UK). Labelling efficiencies were between 75% and 90% for each dye, as determined using the manufacturer's quantification method and assuming a molar extinction coefficient of 70,820 cm$^{-1}$ for SecY$_{MK}$EG.

## Ensemble translocation assays

SecYEG and SecY$_{MK}$EG were reconstituted into proteoliposomes with *E. coli* polar lipids to a final concentration of 4.6 µM and extruded to 400 nm. Classical protease protection translocation assays were performed by mixing PLs (46 nM), creatine kinase (0.1 mg/mL), creatine phosphate (5 mM), SecA (300 nM), preprotein (1 µM) and, for stated reactions, SecB (10 µM), in TKM buffer (20 mM Tris pH 7.5, 50 mM KCl, 2 mM MgCl$_2$) to a final volume of 100 µL. Reactions were preincubated at 25°C for 5 min and initiated by addition of ATP (1 mM). After 30 min at 25°C, 50 µL of reaction was

quenched in 50 µL of ice cold HEPES containing protease K at 0.6 mg/mL. To ensure complete proteolysis of un-translocated preprotein, samples were left to incubate on ice for 20 min. Proteins were then precipitated by addition of trichloroacetic acid to 20 mM and the samples were centrifuged at 15,000 g for 10 min. The supernatant was removed and the pellets were left to dry for two hours in a speed vacuum (LABOGENE). The pellets were resuspended in 10 µL 1x LDS buffer and left overnight. The following morning the samples were analysed by SDS PAGE and immunoblotting, using a C-terminal V5 epitope for detection.

## Ensemble ATPase assays

ATPase assays were conducted using an NADH-based enzyme-linked ATP regeneration system. Reactions were prepared in TKM and contained SecYEG PLs (46 nM), SecA (300 nM), NADH (200 µM), and Pyruvate Kinase/Lactate Dehydrogenase from rabbit muscle (~10 units/mL, Sigma). NADH absorbance at 340 nm was monitored with a Perkin Elmer Lambda 25 spectrophotometer equilibrated at 25°C. After 5 min of equilibration, ATP was added to a final concentration of 1 mM and basal ATPase activity of SecA was observed. 10 min later preprotein was added to a saturating concentration to initiate translocation. ATP hydrolysis rates were calculated from the linear phase of 340 nm absorbance decrease following addition of preprotein, indicative of steady-state SecA ATPase activity.

## SmFRET in msALEX TIRF configuration on immobilised proteoliposomes

SecY$_{MK}$EG was reconstituted into proteoliposomes (PLs) with *E. coli* polar lipid to a final concentration of 1.5 nM and extruded to 100 nm: at this concentration and size, most liposomes are expected to contain either 0 or 1 copy of SecY$_{MK}$EG (*Deville et al., 2011*).

PLs were immobilised on a glass supported lipid bilayer and imaged with a previously described TIRF set-up (*Sharma et al., 2014*) extended with msALEX illumination (*Lee et al., 2005*). The alternation cycle consisted of 100 ms cyan (488 nm) and orange (594 nm) excitation periods, adding information about stoichiometry of dyes and thus allowed to filter out singly labelled molecules.

The buffer used was TKM (20 mM Tris pH 7.5, 50 mM KCl, 2 mM MgCl$_2$) with 1 mM 6-hydroxy-2,5,7,8-tetramethylchroman-2-carboxylic acid (TROLOX) and GODCAT, enzymatic oxygen scavenging system, consisting of a mix of glucose oxidase, catalase, β-D-glucose (*Aitken et al., 2008*) to limit photobleaching. Immobilised samples were supplemented with 1 µM SecA, 10 µM SecB (if present), 700 nM proOmpA (if present), 1 mM ADP or 1 mM AMPPNP or varying concentrations of ATP (unless stated otherwise in figure legend). Translocation assays under saturating ATP conditions were supplemented with an ATP regeneration system (50 µg/ml creatinine kinase, 10 mM phosphocreatine). TIRF movies (200 ms resolution) were taken from samples immediately after mixing of all components directly at the microscope stage.

The data were analyzed in iSMS software (*Preus et al., 2015*). The two channels of each image were aligned and fluorescence count traces (donor and acceptor) were extracted and raw FRET efficiencies (E) and stoichiometries (S) were computed. To eliminate contributions from complexes with single type of dye or photobleached acceptor dye, only traces with S values between 0.25 and 0.75 were selected for further analysis. Another selection criterion for molecules was anti-correlation of intensity in donor and acceptor channels. All trajectories were also checked for bleaching and blinking events. Molecules showing bleaching were used to obtain correction factors, that is donor leakage, direct acceptor excitation and gamma factor (*Preus et al., 2015*). Experiments were repeated at least three times using independent proteoliposome preparations. To construct histograms from TIRF data, we used only such TIRF time traces which showed transitions detectable by Hidden Markov Model algorithm implemented in iSMS (*Preus et al., 2015*), that is were responsive/active. This approach eliminated contribution from SecYEG complexes which were reconstituted with their cytoplasmic side facing the vesicle interior and thus inaccessible to SecA and the substrate. Corrected FRET values were used to produce histograms.

## Photobleaching correction

Statistical distributions of dwell times were corrected for photobleaching using probability distribution of photobleaching times (P$_{photobleaching}$) estimated from TIRF traces for each experiment. Subsequently, we employed non-negative regularised iterative reconvolution of two distributions, that is

photobleaching of individual molecules and simulated dwell time distribution to match measured data:

$$P_{measured} = P_{photobleaching} * P_{estimate},$$ (1)

where $P_{measured}$ relates to the Probability Density Function (PDF) derived from experiment, $P_{photobleaching}$ is PDF of photobleaching (estimated from data) and $P_{estimate}$ is the simulated gamma distribution with scale equal to one, representing an estimate of the original data unaffected by photobleaching. The best hit was found by a least squares method as implemented in SciPy optimize python package (http://www.scipy.org/). A regularised inverse transform was used to reconstruct dwell-time histograms using non-negativity constraints (*Provencher, 1982*) and the reconstructed probability density function, $P_{estimate}$.

## μsALEX confocal experiments on freely diffusing proteoliposomes

The experimental set-up used to collect μsALEX data was previously described (*Sharma et al., 2014*). The laser alternation period was set to 40 μs (duty cycle of 40%) with intensity for the 488 nm laser ~100 μW and the 594 nm laser intensity ~90 μW. Data were collected using Labview graphical environment (LabView 7.1 Professional Development System for Windows, National Instruments, Austin, TX) (*Lee et al., 2005*). Separate photon streams were then converted and stored in an open file format for timestamp-based single-molecule fluorescence experiments (photon-hdf5), which is compatible with many recent data processing environments (*Ingargiola et al., 2016a*).

Fluorescence bursts were analysed by customised python programming scripts (see Supplementary Source Code File: Python source code for data processing and the associated Source Data) based on the open source toolkit for analysis of freely-diffusing single-molecule FRET bursts (*Ingargiola et al., 2016b*). The background was estimated as a function of time, respecting exponentially distributed photon delays generated by a Poisson process. In order to guarantee a maximal signal-to-background ratio, we used background dependent dual-channel burst search (DCBS) (*Michalet et al., 2013*; *Nir et al., 2006*) in sliding window mode (*Eggeling et al., 1998*), which effectively deals with artifacts due to photophysical effects such as blinking. Further filtering was based on dye stoichiometry (S within 0.25–0.75).

Three correction parameters: γ-factor, donor leakage into the acceptor channel and acceptor direct excitation by the donor excitation laser were employed and determined using polyproline standards of different length as FRET samples (*Best et al., 2015*; *Sharma et al., 2014*). Corrections were applied at the population-level (*Lee et al., 2005*) to avoid distortion of the FRET distributions (*Gopich and Szabo, 2007*).

Filtered bursts were then assembled into 2D E-S histograms and 1D probability density function plots were generated using library Seaborn, based on Matplotlib (*Hunter, 2007*). Subpopulations were fitted to weighted Gaussian mixture models using Scikit (*Pedregosa et al., 2011*).

SecY$_{MK}$EG in our study is reconstituted into liposomes; therefore, 50% of all complexes inevitably end up facing the opposite orientation (with cytoplasmic side facing inwards) which is unable to bind SecA and translocate (unresponsive population). FRET distributions derived via RASP analysis were corrected for contribution from the 50% SecY$_{MK}$EG in opposite orientation by subtracting appropriately scaled FRET distribution of SecY$_{MK}$EG alone (see *Figure 2—figure supplement 1A* for one-dimensional histogram and *Figure 4A* for two dimensional transition plot) from the data (see *Figure 2—figure supplement 2* for illustration of the procedure).

## Recurrence analysis of single particles (RASP)

To analyse events on timescales from 100 μs to ~100 ms, we employed RASP which relies on extremely diluted samples, where the probability for a molecule to return to the confocal volume is greater than the probability of a new molecule being detected (*Hoffmann et al., 2011*). RASP extracts time resolved information for FRET subpopulations by constructing recurrence FRET efficiency histograms. These are acquired by first selecting photon bursts from a small transfer efficiency range (initial bursts) and then building the FRET efficiency histogram only from bursts detected within a precisely defined short time interval (the recurrence interval) after all selected initial bursts. Systematic variation of the recurrence interval allows determination of the kinetics of interconversion between subpopulations.

The longest usable recurrence time is related by concentration and diffusion time of the observed objects and can be set based on the recurrence probability. To estimate the recurrence probability of single molecules, we employed a correlative approach (Hoffmann et al., 2011). Bursts from different and non-interacting molecules are expected to be uncorrelated. On the other hand, bursts originating from the same molecule should be correlated and a 'same molecule' probability $P_{same}(\tau)$ was calculated as:

$$P_{same}(\tau) = 1 - 1/g(\tau) \tag{2}$$

where $g(\tau)$ is the burst time autocorrelation function of all detected bursts. From a fit to the data, we determined for each burst pair the probability that it originated from the same, recurring molecule, and calculated the average $P_{same}$ for a subset of bursts by averaging over all corresponding burst pairs.

## Recurrence transfer efficiency histograms

To derive kinetics from RASP, we constructed transfer efficiency histograms from a set of bursts selected by two criteria. First, the bursts $b_2$ must be detected during a time interval between $t_1$ and $t_2$ (the 'recurrence interval', $T = (t_1, t_2)$) after a previous burst $b_1$ (the 'initial burst'). Second, the initial bursts must yield a transfer efficiency, $E(b_1)$, within a defined range, $\Delta E_1$ (the 'initial E range'). The set R of burst pairs $\{b_1, b_2\}$ selected by these criteria is then:

$$R(\Delta E_1, T) = \{\{b_1, b_2\} \,|\, E(b_1) \in \Delta E_1, tb_2 - tb_1 \in T\} \tag{3}$$

where $tb_1$ and $tb_2$ are the detection times of the bursts $b_1$ and $tb_2$, respectively. The set of burst pairs R is the starting point for the different types of analysis presented here. A very informative way of representing the data is the FRET efficiency histogram of all values $E(b_2)$, the 'recurrence transfer efficiency histogram'.

## Cross-peaks in 2D recurrence transfer efficiency contour plots

As a visual guide, we constructed 2D transition density contour plots. They were obtained from two-dimensional Gaussian KDE analysis (*Scott, 1992*) of burst pairs, where the initial burst and the second burst yield transfer efficiencies in range $\Delta E1$ and $\Delta E_2$, respectively. Each plot was constructed for a certain recurrence interval T.

2D contour plots were also used to address a common issue in the analysis of transfer efficiency histograms: the determination of the number of contributing subpopulations and their peak shapes. These were answered by choosing short recurrence intervals and initial transfer efficiency ranges that represent only a single subpopulation. The significance of small populations and the properties of strongly overlapping peaks were tested with this approach.

## Interconversion dynamics from kinetic recurrence analysis

To extract rates of interconversion between subpopulations from time-dependent recurrence E histograms, we constructed histograms for different recurrence intervals, and extracted the fraction of a subpopulation versus time. To determine the rates of interconversion, we related the change in the recurrence E histograms with increasing recurrence times to the dynamics of the interconversion process as was first shown by (*Hoffmann et al., 2011*).

For a system populating two states A and B, we defined the probability pA ($\tau, \Delta E_1$) that from the set of burst pairs R($\Delta E_1$,T), b2 originates from a molecule in state A. pA ($\tau, \Delta E_1$) was determined from global fitting the corresponding recurrence histogram and determining the ratio of the peak area corresponding to subpopulation A over the total area under the peaks corresponding to A and B (for details see [*Hoffmann et al., 2011*]).

$$pA(\tau, \Delta E1) = p_{same}(\tau) \, p_A^{i=j}(\tau, \Delta E1) + \left[1 - p_{same}(\tau)\right] p_A^{i \neq j} \tag{4}$$

where $p_A^{i=j}(\tau, \Delta E1)$ denotes the probability that a recurring molecule ($i = j$) is in state A, and $p_A^{i \neq j}$ is the probability that a newly arriving molecule ($i \neq j$) (leading to burst $b_2$) is in state A. $p_A^{i \neq j}$ is probability of measuring a burst originating from a molecule in state A and it was determined from globally

fitted areas under the corresponding peak functions extracted from a set of transfer efficiency histograms.

The time dependence of $p_A^{i=j}$ is determined by the interconversion kinetics between states A and B and is defined as:

$$p_A^{i=j}(\tau, \Delta E1) = \left(1 + \varepsilon \left(\frac{1}{\rho A(\tau, \Delta E_1)} - 1\right)\right)^{-1} \tag{5}$$

with

$$\rho A(\tau, \Delta E_1) = \rho_A^{eq} + \left[\rho A(0, \Delta E1) - \rho_A^{eq}\right] e^{-\lambda\tau}, \tag{6}$$

Here, $\rho_A(\tau, \Delta E_1)$ is the probability that a protein that emitted a burst at time 0 with a transfer efficiency in the range $\Delta E_1$ is in state $A$ at time $\tau$. $\rho_A(0, \Delta E_1)$ and $\rho_A^{eq}$ are the corresponding initial and equilibrium probabilities, respectively. The kinetic rate constant $\lambda$ corresponds to the sum of the forward and backward rate constants of interconversion between A and B.

## Burst variance analysis (BVA)

The BVA method is capable to identify dynamics in FRET distributions [1, 2]. The main idea of BVA is to subdivide bursts into contiguous sub-bursts consisting of a fixed number of photons (n), and to compare the variance of acceptor photons of all sub-bursts within each burst. This is later compared to the theoretical shot-noise-limited variance. An empirical variance of sub-bursts larger than the shot-noise-limited variance for a certain FRET region indicates the presence of dynamics.

In a FRET subpopulation originating from static biomolecules, the sub-burst acceptor counts $n_a$ have a binomial distribution, $N_a \sim B(n, E_p)$, where n is the number of photons in each sub-burst and $E_p$ is the estimated population proximity-ratio.

If $N_a$ follows a binomial distribution, the random variable $E_{sub} = N_a/n$, has a standard deviation reported in *Equation (7)*.

$$S_{E_{sub}} = \left(\frac{E_p(1 - E_p)}{n}\right)^{1/2} \tag{7}$$

In detail, BVA analysis comprises of four major steps: (1) division of individual bursts into consecutive sub-bursts containing a constant number of consecutive photons n, (2) computation of the $E_p$ of all sub-bursts, (3) computation of the empirical standard deviation ($s_E$) of sub-bursts $E_p$ in each burst, and (4) comparison of $s_E$ to the expected standard deviation of a shot-noise-limited distribution for a given mean $E_p$ (see *Equation 7*).

If the detected FRET efficiency distribution comes from a static mixture of sub-populations characterized by distinguishable FRET efficiencies, $s_E$ of each burst is only affected by shot-noise and will follow the analytical function shown in *Equation (7)*.

On the other hand, if the observed distribution comes from molecules belonging to a single species, undergoing transition between different FRET states (over the timescale comparable to the diffusion time of molecular species of interest), $s_E$ of each burst will be larger than the expected shot-noise-limited standard deviation, and it will appear above the shot-noise standard deviation curve. Other, and preferred way, to distinguish between static and dynamic subpopulations is to compute confidence intervals using Monte Carlo algorithm described below.

To calculate upper-limit confidence intervals on $s_E$, we need to consider the sampling distribution of standard deviations, $P(s_E)$, expected for M windows of n photons. To implement the Monte Carlo approach, we simulate the sampling distribution of $s_{shot\_noise}$,

$$S_{shot\_noise} = \sqrt{\sum_{\substack{i\,where \\ L \leq PR_i < U}} \sum_{j=1}^{M_i} \left[\frac{\left(\frac{F_A^{ij}}{n} - \mu\right)^2}{\sum M_i}\right]} \tag{8}$$

where

$$\mu = \sum_{\substack{i\,where \\ L \leq PR_i < U}} \sum_{j=1}^{M_i} \left( \frac{\frac{F_A^{ij}}{n}}{\sum M_i} \right) \tag{9}$$

$F_A^{ij}$ are random variables drawn from a binomial distribution with $n$ trials (i.e. the number of photons per each window) and $PR$ is a probability of success. We define the resulting Monte Carlo distribution as $P_{MC}(s_{\text{shot\_noise}})$.

Then we use the $P_{MC}(s_{\text{shot\_nouse}})$ to calculate the upper-tail confidence interval on the standard deviation, $S_{shot\_noise}^{CI}$, and test for dynamics by comparing it to the observed $s_E$. Unless otherwise indicated, per-experiment confidence levels were set to $\alpha = .001$; deviations beyond the value of $s_E$ corresponding to this level should reflect the presence of dynamics.

It is important to note that we use the PR here because, regardless of the real FRET efficiency, the detected counts are partitioned between donor and acceptor channels according to a binomial distribution with success probability equal to the PR. If we used corrected FRET efficiency, calculation of variance would be biased.

## Acknowledgements

This paper is dedicated to our friend and colleague, the late Prof. Steve Baldwin. This work was funded by the BBSRC: TF, RT and SER (BB/N017307/1), DW and IC (BBSRC: BB/N015126/1 and BB/I008675/1), PO, RT, SER and SAB (BB/I008675/1), JEH (BB/M011151/1), IC and WJA (BB/I006737/1); RAC (BBSRC South West Bioscience Doctoral Training Partnership and BB/M003604/I). Additional support was provided by the Wellcome Trust (104632) to IC and WJA and ERC ((FP7/2007-2013)/ ERC grant agreement 32240) to SER. TF and RT are supported from European Regional Development Fund-Project 'Mechanisms and dynamics of macromolecular complexes: from single molecules to cells' (No. CZ.02.1.01/0.0/0.0/15_003/0000441). We thank Dr. Marek Scholz for help with Python code development.

## Additional information

### Funding

| Funder | Grant reference number | Author |
|---|---|---|
| Biotechnology and Biological Sciences Research Council | BB/N017307/1 | Tomas Fessl<br>Sheena E Radford<br>Roman Tuma |
| Wellcome | 104632 | William John Allen<br>Ian Collinson |
| Seventh Framework Programme | 32240 | Sheena E Radford |
| European Regional Development Fund | CZ.02.1.01/0.0/0.0/15_003/0000441 | Tomas Fessl<br>Roman Tuma |
| Biotechnology and Biological Sciences Research Council | BB/N015126/1 | Daniel Watkins<br>Ian Collinson |
| Biotechnology and Biological Sciences Research Council | BB/I008675/1 | Daniel Watkins<br>Peter Oatley<br>Steve A Baldwin<br>Sheena E Radford<br>Roman Tuma |
| Biotechnology and Biological Sciences Research Council | BB/M011151/1 | Jim Horne |
| Biotechnology and Biological Sciences Research Council | BB/I006737/1 | William John Allen<br>Ian Collinson |
| Biotechnology and Biological Sciences Research Council | BBSRC South West Bioscience Doctoral Training Partnership | Robin Adam Corey |

| Biotechnology and Biological Sciences Research Council | BB/M003604/I | Robin Adam Corey |

The funders had no role in study design, data collection and interpretation, or the decision to submit the work for publication.

## Author contributions

Tomas Fessl, Conceptualization, Data curation, Software, Formal analysis, Investigation, Visualization, Methodology, Writing—original draft, Writing—review and editing; Daniel Watkins, Conceptualization, Resources, Data curation, Formal analysis, Validation, Visualization, Writing—original draft, Writing—review and editing; Peter Oatley, Resources, Data curation, Formal analysis, Investigation, Visualization, Methodology, Writing—review and editing; William John Allen, Conceptualization, Resources, Formal analysis, Validation, Methodology, Writing—original draft, Writing—review and editing; Robin Adam Corey, Resources, Data curation, Formal analysis, Validation, Investigation, Writing—review and editing; Jim Horne, Resources, Writing—review and editing; Steve A Baldwin, Conceptualization, Supervision, Funding acquisition, Project administration; Sheena E Radford, Conceptualization, Formal analysis, Supervision, Funding acquisition, Writing—original draft, Project administration, Writing—review and editing; Ian Collinson, Conceptualization, Resources, Formal analysis, Supervision, Funding acquisition, Validation, Writing—original draft, Project administration, Writing—review and editing; Roman Tuma, Conceptualization, Data curation, Software, Formal analysis, Supervision, Funding acquisition, Visualization, Methodology, Writing—original draft, Project administration, Writing—review and editing

## Author ORCIDs

Tomas Fessl (iD) http://orcid.org/0000-0001-6969-4870
Daniel Watkins (iD) http://orcid.org/0000-0003-3825-5036
Peter Oatley (iD) http://orcid.org/0000-0001-7881-5590
William John Allen (iD) http://orcid.org/0000-0002-9513-4786
Robin Adam Corey (iD) http://orcid.org/0000-0003-1820-7993
Jim Horne (iD) http://orcid.org/0000-0001-5260-2634
Sheena E Radford (iD) http://orcid.org/0000-0002-3079-8039
Ian Collinson (iD) https://orcid.org/0000-0002-3931-0503
Roman Tuma (iD) http://orcid.org/0000-0003-0047-0013

## Decision letter and Author response

Decision letter https://doi.org/10.7554/eLife.35112.029
Author response https://doi.org/10.7554/eLife.35112.030

# Additional files

## Supplementary files

• Source code 1. Python script source code.
DOI: https://doi.org/10.7554/eLife.35112.025

• Source data 1. Primary single molecule fluorescence data file for use with the Python script.
DOI: https://doi.org/10.7554/eLife.35112.026

• Transparent reporting form
DOI: https://doi.org/10.7554/eLife.35112.027

## Data availability

Compressed data are available together with the relevant scripts as supplementary source data and code.

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
