## [Decision Letter]

Thank you for submitting your article "Dynamic action of the Sec machinery during initiation, protein translocation and termination" for consideration by *eLife*. Your article has been reviewed by three peer reviewers, and the evaluation has been overseen by a Reviewing Editor and Richard Aldrich as the Senior Editor. The following individuals involved in review of your submission have agreed to reveal their identity: Aaron Lucius (Reviewer #1); Franck Van Hoa Duong (Reviewer #3).

The reviewers have discussed the reviews with one another and the Reviewing Editor has drafted this decision to help you prepare a revised submission.

Summary:

This manuscript seeks to examine the mechanism of translocation catalyzed by SecYEG. Specifically, the manuscript reports results from single molecule experiments designed to examine SecY pore dynamics. Through careful placement of the donor and acceptor dyes the observed FRET signal is sensitive to opening and closing of the channel. Overall, this manuscript represents a significant step toward a better understanding of translocon dynamics using an innovative single-molecule approach. The observation of plug dynamics helps to clarify how SecA, ATP, the signal sequence, and the substrate protein regulate SecYEG channel activity. The translocation and initiation rates determined here are important parameters for understanding how the SecA motor functions. The study clearly illustrates the power of single molecule measurements which, when carefully executed as done here, bring valuable quantitative insights to complex reaction mechanisms.

Essential revisions:

1) The finding that apo SecYEG appears to populate the open state about half of the time is surprising. As the authors state at the end of the Results section, opening of the channel without translocation "would dissipate the PMF for no obvious gain". Yet, from the FRET histograms, it looks like the open state is populated with a probability of ~0.5, which appears to contradict electrophysiology experiments showing that the resting channel does not conduct. The authors should do similar measurements comparing conductivities of resting wild-type SecYEG and their labeled SecYEG mutant protein to rule out that introduction of the dyes perturb the structural equilibrium.

2) The 0.25 FRET state in the presence of SS is interpreted as the unlocked state of the translocon. It is not clear if this is correct, or it actually is a state. How is it ruled out that the FRET value of 0.25 represents an average of open and closed, similar to the 0.35 FRET state of SecYEG alone? SS binding may just shift the equilibrium, not result in the population of a different state.

3) When assigning FRET populations to functional states – a key experiment for the paper – authors report a FRET value lower than expected for the closed state (~0.35 vs 0.4). This lower than expected FRET value was observed with SecYEG in the absence of both SecA and substrate. In contrast, the FRET value reported in the presence of SecA, ATP and no substrate is ~0.4. From this, they conclude that plug is intrinsically unstable in the SecYEG complex in the absence of SecA, but authors do not provide direct evidence for the statement. Instead, authors use ATP and SecA without pOA, to conclude that the closed state is only attained within the SecY_MK_EG:SecA complex. We wonder why authors did not measure dwelling time after introducing sec or Prl mutations into SecY or SecE which reportedly impact on the plug domain mobility. Of particular interest is the SecY357 mutant, in which the plug is largely in the closed state, or SecY PrlA4, in which the plug is biased toward the open state. As it is, one may question whether the SecA interaction on the cytosolic side of SecY is rather perturbing FRET efficiency than plug dynamics. The window of FRET efficiency is quite narrow (0.2 vs 0.4 and value in between), as such unambiguous assignment seems critical for this as well as future smFRET based work. At any rate, discussion commentaries on the above points should be presented to the readers; it is not immediately obvious why SecA would stabilize the translocon closed state.

---

## [Author Response]

Essential revisions:1) The finding that apo SecYEG appears to populate the open state about half of the time is surprising. As the authors state at the end of the Results section, opening of the channel without translocation "would dissipate the PMF for no obvious gain". Yet, from the FRET histograms, it looks like the open state is populated with a probability of ~0.5, which appears to contradict electrophysiology experiments showing that the resting channel does not conduct. The authors should do similar measurements comparing conductivities of resting wild-type SecYEG and their labeled SecYEG mutant protein to rule out that introduction of the dyes perturb the structural equilibrium.

Thank you for this suggestion and we apologize that our manuscript did not make this clear. Indeed, further analysis and experiments show that SecYEG is rarely open, consistent with previous electrophysiology experiments, as delineated in detail below.

The open state is only significantly populated during translocation, i.e. under steady-state conditions with multiple turnovers, as seen in Figure 2D and F, from which the respective histograms in panels E and G were computed. In other words, with SecYEG alone the plug wobbles around a partly open state, but never stays in the fully open state for significant amount of time. In addition, SecYEG contains a 6 residue hydrophobic ring, which also maintains the channel sealed in the absence of pre-protein. This is now clarified in subsection “Translocon unlocking by the signal sequence is necessary, but not sufficient, for plug opening” and by Figure 2 legend explicitly stating that data were collected under steady state translocation conditions.

We have also clarified the resting state ensemble of the SecYEG translocon alone (Figure 2—figure supplement 1) and its relation to the open state observed during translocation. Our observations are not compatible with prolonged opening and leakage of the channel in the resting state. Because of inherent shot noise the broad peak in Figure 2—figure supplement 1A may suggest very moderate population within the open region around E=0.2. However, when the broad peak is deconstructed into the expected contributions from the closed (as seen in the SecYEG:SecA: 2 mM ATP sample; Figure 2—figure supplement 1B), open (as seen in the SecYEG:SecA:pOA:ATP sample quenched during translocation with AMP-PNP) and a middle peak accounting for the dynamic state (as shown in the revised Figure 2—figure supplement 1A yellow), it is clear that the open state (dashed green line in the revised Figure 2—figure supplement 1A) is seldom populated (< 7% ).

This is further corroborated by the transition FRET diagram in Figure 4A, where the absence of density at the open state region (E1 = E2 ~0.2) indicates that even if the complex visits the open state basin (see density E1 = 0.2 and E2 = 0.3) it leaves it within 21 ms, the time window for compiling these transition diagrams. Such transient openings are unlikely to cause significant proton leakage, especially given the additional protection of the central hydrophobic ring.

On the other hand, the dynamic, partly open state of the plug is consistent with previous crosslinking experiments reported by Tam et al. EMBO J. 2005 and the restricted conformational heterogeneity of the R357E mutant (Figure 4—figure supplement 1), which stabilizes the plug in the closed state (see also response to essential revision 3 below). We have added this reasoning to the revised text in the Results section, where we also discuss the extent the dyes may influence the plug dynamics. There, we have also added a conductivity-related reference.

2) The 0.25 FRET state in the presence of SS is interpreted as the unlocked state of the translocon. It is not clear if this is correct, or it actually is a state. How is it ruled out that the FRET value of 0.25 represents an average of open and closed, similar to the 0.35 FRET state of SecYEG alone? SS binding may just shift the equilibrium, not result in the population of a different state.

We have now performed time-resolved analyses to examine further the possibility of rapid interconversion between states for the SS unlocked states in Figure 4G-H. With fast time resolution (0.1 ms) Burst Variance Analysis (BVA, Figure 4—figure supplement 2) does not exhibit any indication of dynamics. Of course, the plug may exhibit fluctuations on the ns timescale but this is unlikely since spontaneous closing of the plug (Figure 3D) takes place on slower, millisecond time scale. We have added a new Figure 4—figure supplement 2 with BVA and a new paragraph in subsection “Translocon unlocking by the signal sequence is necessary, but not sufficient, for plug opening”.

3) When assigning FRET populations to functional states – a key experiment for the paper – authors report a FRET value lower than expected for the closed state (~0.35 vs 0.4). This lower than expected FRET value was observed with SecYEG in the absence of both SecA and substrate. In contrast, the FRET value reported in the presence of SecA, ATP and no substrate is ~0.4. From this, they conclude that plug is intrinsically unstable in the SecYEG complex in the absence of SecA, but authors do not provide direct evidence for the statement. Instead, authors use ATP and SecA without pOA, to conclude that the closed state is only attained within the SecY_MK_EG:SecA complex. We wonder why authors did not measure dwelling time after introducing sec or Prl mutations into SecY or SecE which reportedly impact on the plug domain mobility. Of particular interest is the SecY357 mutant, in which the plug is largely in the closed state, or SecY PrlA4, in which the plug is biased toward the open state. As it is, one may question whether the SecA interaction on the cytosolic side of SecY is rather perturbing FRET efficiency than plug dynamics. The window of FRET efficiency is quite narrow (0.2 vs 0.4 and value in between), as such unambiguous assignment seems critical for this as well as future smFRET based work. At any rate, discussion commentaries on the above points should be presented to the readers; it is not immediately obvious why SecA would stabilize the translocon closed state.

The use of these mutants was indeed an excellent idea and we are grateful to the reviewer(s) for suggesting it. We tried to pursue both the “closed” R357E mutant and the PrlA4 “open” destabilized plug variant. While we were successful in purifying and labelling the former, expression of the latter did not yield a stable protein, presumably due to introducing two mutations (Cys necessary for labelling) in the already dynamic plug domain. However, successful characterization of the labelled SecY_MK,R357E_ variant demonstrates that the plug preferentially assumes the closed, high FRET position and fails to open upon activation, in accord with previous biochemical data of Duong and colleagues. In addition, the single molecule results for SecY_MK_EG (wild type background) and SecY_MK,R357E_EG support previous cross-linking results in which the plug of the wild type complex can be crosslinked to the open position while the mutant resists. This suggests that the plug in SecYEG alone is indeed dynamic (wobbly), and SecA binding limits the mobility allosterically (see revised text, in subsection “Translocon unlocking by the signal sequence is necessary, but not sufficient, for plug opening”). Furthermore, given the almost complete absence of SecY dimers in single molecule experiments (due to extreme dilution during reconstitution) the R357E mutation must have additional effects over and above dimer destabilization (as known from the literature), for example allosteric stabilisation of the plug within individual monomers. We have added this interpretation to subsection “Translocon unlocking by the signal sequence is necessary, but not sufficient, for plug opening”.